# Unsupervised Domain Adaptation via Minimized Joint Error

## Abstract

Unsupervised domain adaptation transfers knowledge from a learned source domain to a different target distribution, for which only few or no labeled data is available. Some researchers proposed upper bounds for the target error when transferring the knowledge, i.e., Ben-David et al. (2010) established a theory based on minimizing the source error and distance between marginal distributions simultaneously. However, in most works the joint error is usually ignored due to the intractability. In this paper, we argue that the joint error is essential for the domain adaptation problem, in particular if the samples from different classes in source/target are closely aligned when matching the marginal distributions due to a large domain gap. To tackle this problem, we propose a novel objective that relates to an upper bound of the joint error. Moreover, we adopt a source/pseudo-target labels induced hypothesis space that can reduce the searching space to further tighten up this bound. For the dissimilarity measurement between hypotheses, we propose a novel cross margin discrepancy to alleviate the instability during adversarial learning. In addition, we present extensive empirical evidence that shows that our proposal boosts the performance in image classification accuracy on standard domain adaptation benchmarks.

## 1 Introduction

The advent of deep neural network (Krizhevsky et al., 2012) brings visual learning into a new era. However, the performance heavily relies on the abundance of annotated data. Since traditional machine learning assumes a model is trained and verified in a fixed distribution, it cannot always be applied to real-world problems directly. Take image classification as an example, a number of factors, such as the change of light, noise, angle in which the image is pictured, and different types of sensors, can lead to a domain gap that harms the performance when predicting on test data. In many practical cases, a model trained in one or more source domains is desired that is also applicable to another domain. As a solution, domain adaptation (DA) aims to transfer the knowledge learned from a source distribution, which is typically fully labeled, into a different (but related) target distribution. This work focus on the most challenging case, i.e., unsupervised domain adaptation (UDA), where no target label is available.

Ben-David et al. (2010) suggest that the target error can be minimized by bounding the error on source data, the discrepancy between distributions of the two domains, and a small optimal joint error. Therefore, many researchers focus on learning domain-invariant features such that the discrepancy can be minimized. For aligning distributions across domains, mainly two strategies have been substantially explored. The first one bridges the distributions by matching their statistics (Long et al., 2015; 2017; Pan et al., 2009). The second strategy utilizes adversarial learning (Goodfellow et al., 2014) to build a minimax game, where a domain discriminator is trained to distinguish the source from the target while the feature extractor is trained to confuse it simultaneously (Ganin & Lempitsky, 2015; Ganin et al., 2016; Tzeng et al., 2017). Despite the remarkable results achieved by distribution matching schemes, they still suffer from a major limitation: the joint distributions of feature spaces and categories are not well aligned across domains. As reported in Ganin et al. (2016), such methods fail to generalize in closely related source/target pairs, e.g., the adaptation from MNIST to SVHN. One potential reason is when matching marginal distributions between two domains,

samples from different classes can get mixed together. In this case, the joint error becomes non-negligible since no hypothesis can jointly classify source and target data with a high accuracy.

This work aims to address the above problem by incorporating the joint error to formalize an optimizable upper bound such that the undesired overlap due to a wrong match can be properly penalized. UDA classification problems can be perfectly solved if we can match the conditional distributions of each class for source and target domain. In practice, however, most of the existing methods choose to deal with marginal distributions as we lack target labels (Ganin et al., 2016; Tzeng et al., 2017; Saito et al., 2017b; Zhang et al., 2019b; Kim et al., 2019). Motivated by this, we come up with a specially designed distance measurement, namely cross margin discrepancy, that leads our proposed upper bound to an objective that resembles CGAN [1] (a way to match conditional distributions) (Mirza & Osindero, 2014). We evaluate our proposal on several different classification tasks. In most experimental settings, our method improves the performance by a large margin. The contributions of this work can be summarized as follows:

· We propose a novel objective that relates to an upper bound of the joint error and show that our proposal can reduce the chance of misalignment through the marginal distribution matching.

· We construct a specific hypothesis space induced by source/pseudo-target labels to tighten up the proposed objective so that we can avoid optimizing a loose bound within an immense searching space.

· We adopt a novel measurement, namely cross margin discrepancy, that measures the dissimilarity between hypotheses. We show that using this novel measurement, we can alleviate the instability during adversarial learning.

· We present extensive empirical evidence showing that our proposal gives more reliable performance in image classification on several domain adaptation benchmarks compared with other upper bound related methods.

## 2 Related Work

The upper bound proposed by Ben-David et al. (2010) invoked numerous approaches focusing on reducing the gap between source and target domains by learning domain-invariant features, which can be achieved through statistical moment matching. Maximum Mean Discrepancy (MMD) (Long et al., 2015; 2017) is used to match the hidden representations of certain layers in a deep neural network. Transfer Component Analysis (TCA) (Pan et al., 2011) tries to learn a subspace across domains in a Reproducing Kernel Hilbert Space (RKHS) using MMD that dramatically minimizes the distance between domain distributions. Adaptive batch normalization (AdaBN) (Li et al., 2018) modulates the statistics from source to target on batch normalization layers across the network in a parameter-free way.

Another way to learn domain-invariant features is by leveraging generative adversarial network to produce target features that exactly match the source. Ganin & Lempitsky (2015) relax divergence measurement in the upper bound by a worst case which is equivalent to the maximum accuracy that a discriminator can possibly achieve when distinguishing source from target. Tzeng et al. (2017) follow this idea but separate the training procedure into classification stage and adversarial learning stage where an independent feature extractor is used for target. Saito et al. (2017b) explore a tighter bound by explicitly utilizing task-specific classifiers as discriminators such that features near the support of source samples will be favored by extractor. Zhang et al. (2019b) introduce margin disparity discrepancy, a novel measurement with rigorous generalization bounds, tailored to the distribution comparison with the asymmetric margin loss to bridge the gap between theory and algorithm. Methods that perform distribution alignment on pixel-level in raw input, which is known as image-to-image translation, are also proposed (Liu & Tuzel, 2016; Bousmalis et al., 2017; Sankaranarayanan et al., 2017; Shrivastava et al., 2016; Hoffman et al., 2018; Murez et al., 2017).

Distribution matching can not only bring the source and target domains closer, but also mix samples from different classes together. Therefore, Saito et al. (2017a); Sener et al. (2016); Zhang et al. (2018) aim to use pseudo-labels to learn target discriminative representations which encourage a low-density separation between

---

[1]In Eq.30, we show that our proposal includes a CGAN objective that aligns three pairs of hypothesis induced distributions

classes in the target domain (Lee, 2013). However, this usually requires auxiliary data-dependent hyperparameters to set a threshold for reliable predictions. Long et al. (2018) present conditional adversarial domain adaptation, a principled framework that conditions adversarial adaptation models on the discriminative information conveyed in classifier predictions, where the back-propagation of training objective is highly dependent on pseudo predictions. Wu et al. (2019) raise attention to the arbitrary increase in the joint error caused by distribution matching and propose a relaxed match by restricting the power of domain classifier to deal with the label shifting problem. However, it requires an overlap between the source and target domain in the input space and does not necessarily reduce the joint error. To the best of our knowledge, there is still no research focusing on directly minimizing the joint error for UDA.

In summary, the upper bound proposed by Ben-David et al. (2010) and its extensions (Ganin et al., 2016; Tzeng et al., 2017; Saito et al., 2017b; Zhang et al., 2019b) keep making progress in aligning the marginal distributions between the source and target domains. Nevertheless, the problem of ignoring the joint error remains unsolved. In the following sections, we will theoretically explain how this can harm the performance of the distribution matching in domain adaptation and why our proposal can tackle the problem.

## 3    Proposed Method

In this section, we present the details of our proposal. Firstly, in Sec.3.1 we give our base upper bound and show how it relates to the joint error. Then we provide more specific reasons on the necessity of the joint error theoretically. Secondly, in Sec.3.2 we explain how different choices of hypothesis spaces can affect the optimization of an objective. Finally, in Sec.3.4 we propose a novel measurement for the dissimilarity between hypotheses with a theoretical interpretation and demonstrate its utility in adversarial learning.

### 3.1    An Upper Bound Incorporating Joint Error

We consider the unsupervised domain adaptation as a binary classification task (our proposal holds for multi-class case) where the learning algorithm has access to a set of $n$ labeled points $\{(x_s^i, y_s^i) \in (X \times Y)\}_{i=1}^n$ sampled i.i.d. from the source domain $S$ and a set of $m$ unlabeled points $\{(x_t^i) \in X\}_{i=1}^m$ sampled i.i.d. from the target domain $T$. Let $f_S : X \to \{0, 1\}$ and $f_T : X \to \{0, 1\}$ be the optimal labeling functions on the source and target domains, respectively. Let $\epsilon$ (usually 0-1 loss) denote a distance metric between two functions over a distribution that satisfies the symmetry and triangle inequality. As a commonly used notation, the source error of a hypothesis $h : X \to \{0, 1\}$ in space $H$ is the disagreement w.r.t. the true labeling function $f_S$ under domain $S$, i.e., $\epsilon_S(h) := \epsilon_S(h, f_S)$. Similarly, we use $\epsilon_T(h)$ to represent the error of the target domain. With these notations, we propose the following upper bound for target error:

$$\epsilon_T(h) = \epsilon_T(h, f_T) + [\epsilon_T(h, f_S) - \epsilon_T(h, f_S)] + [\epsilon_S(h, f_S) - \epsilon_S(h, f_S)] + [\epsilon_S(h, f_T) - \epsilon_S(h, f_T)]$$
$$= \epsilon_S(h, f_S) + [\epsilon_T(h, f_T) - \epsilon_T(h, f_S)] + [\epsilon_S(h, f_T) - \epsilon_S(h, f_S)] + \epsilon_T(h, f_S) - \epsilon_S(h, f_T)$$
$$\leq \epsilon_S(h) + \epsilon_T(f_S, f_T) + \epsilon_S(f_S, f_T) + \epsilon_T(h, f_S) - \epsilon_S(h, f_T)$$
$$= \epsilon_S(h) + C_{S,T}(f_S, f_T, h) \tag{1}$$

For simplicity, we use $C_{S,T}(f_S, f_T, h)$ to denote $\epsilon_T(f_S, f_T) + \epsilon_S(f_S, f_T) + \epsilon_T(h, f_S) - \epsilon_S(h, f_T)$. The above upper bound is minimized when $h = f_S$, and it is equivalent to $\epsilon_T(f_S, f_T)$ owing to the triangle inequality:

$$\epsilon_S(h) + \epsilon_S(f_S, f_T) = \epsilon_S(h, f_S) + \epsilon_S(f_S, f_T) \geq \epsilon_S(h, f_T) \tag{2}$$

Furthermore, we demonstrate in such case, our proposal is equivalent to an upper bound of the optimal joint error $\lambda$:

$$\epsilon_T(f_S, f_T) = \epsilon_T(f_S, f_T) + \epsilon_S(f_S, f_S) = \epsilon_T(f_S) + \epsilon_S(f_S) \geq \min_{h \in H}(\epsilon_T(h) + \epsilon_S(h)) = \lambda \tag{3}$$

The reason we focus on the joint error is that Ben-David et al. (2010) state the target error can be upper bounded by the sum of the source risk, the marginal discrepancy between the source and target distributions and the joint error:

$$\epsilon_T(h) \leq \epsilon_S(h) + \sup_{f_1, f_2 \in H} |\epsilon_S(f_1, f_2) - \epsilon_T(f_1, f_2)| + \lambda \tag{4}$$

but plenty of works (Ganin et al., 2016; Tzeng et al., 2017; Saito et al., 2017b; Zhang et al., 2019b; Kim et al., 2019) choose to ignore $\lambda$ and mainly focus on minimizing the marginal discrepancy within a transformed input space $S_g = \{(g(x_s), y_s)|(x_s, y_s) \sim S\}, T_g = \{g(x_t)|x_t \sim T\}$ introduced by a feature extractor $g$:

$$\min_{h,g:h\circ g \in H}[\epsilon_{S_g}(h) + \max_{f_1,f_2:f_1\circ g,f_2\circ g \in H} |\epsilon_{S_g}(f_1, f_2) - \epsilon_{T_g}(f_1, f_2)|] + \lambda_g \qquad (5)$$

where the joint error $\lambda_g = \min_{h^*:h^*\circ g \in H} \epsilon_{S_g}(h^*) + \epsilon_{T_g}(h^*)$. In the original theory (Ben-David et al., 2010), the joint error is indeed an independent term. However, when we introduce the feature extractor $g$ to minimize the marginal discrepancy, we also break the independence. The joint error varies along with the change of $g$, and it can become non-negligible if samples from different classes in the source and target domain get mixed together especially when a huge domain gap exists (Zhao et al., 2019). In that case, no matter how we minimize the marginal discrepancy, the target error is unbounded as not a single $h$ can jointly classify both domains.

In Fig. 1b, we illustrate a case where common methods fail to penalize the undesirable situation. During the distribution matching it can happen that distributions of different classes get aligned together. While this does reduce the marginal discrepancy, it introduces a huge joint error. This can be seen in the right part of Fig. 1b, while the marginal discrepancy (areas 1,3,4 and 6) is relatively small, there is a large joint error (areas 2 and 5). In this case, reducing marginal discrepancy (areas 1,3,4 and 6) leads to an increasing joint error (areas 2 and 5). For the case shown in Fig. 1b, we assume $f_S$ takes a specific form. As a result, $\epsilon_T(f_S, f_T)$ exactly measures the overlapping areas 2 and 5, which is equivalent to the optimal joint error. As the two areas represent the overlaps of different classes, there is no hypothesis that can correctly classify the samples in those areas, which leads to a non-negligible joint error. Our proposal is closely related to the optimal joint error and can adequately reduce the size of overlapping during the distribution matching. Additionally, we provide a Rademacher complexity bound in A.6 according to Mansour et al. (2009).

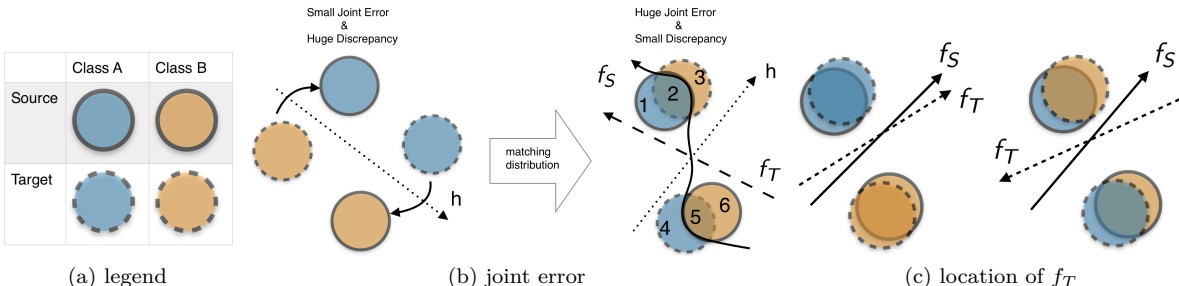

| | | |
|---|---|---|
| (a) legend | (b) joint error | (c) location of $f_T$ |

Figure 1: The left side of a classifier (arrow) is class A and the right side is class B; (a) Legend; (b) Joint error (areas 2 and 5) is penalized such that the feature extractor must try to avoid such an overlap; (c) even with the marginal distribution aligned $f_S$ and $f_T$ might be largely different.

## 3.2 Hypothesis Space Constraint

Since the optimal labeling functions $f_S, f_T$ are not available during the training, we shall further relax the upper bound by taking supremum w.r.t $f_S, f_T$ within a hypothesis space $H^2$:

$$\epsilon_T(h) \le \epsilon_S(h) + C_{S,T}(f_S, f_T, h) \le \epsilon_S(h) + \sup_{f_1,f_2 \in H} C_{S,T}(f_1, f_2, h) \qquad (6)$$

Then minimizing the upper bound of the target error $\epsilon_T(h)$ becomes optimizing a minmax game. However, as the max-player with two parameters $f_1, f_2 \in H$ is much stronger due to a larger searching space, we introduce a feature extractor $g \in H_g$ and transformed feature space $S_g = \{(g(x_s), y_s)|(x_s, y_s) \sim S\}, T_g = \{g(x_t)|x_t \sim T\}$ for the min-player to compensate for it (Now $f_1, f_2 \in H_f \le H$ because $f_{1or2} \circ g \in H$). Applying $g$ to both

---

[2]Our proposal holds even if $f_S, f_T \notin H$, which is proved in A.3

domains, the overall optimization problem can be written as:

$$\min_{g \in H_g, h \in H_f} (\epsilon_{S_g}(h) + \max_{f_1, f_2 \in H_f} C_{S_g, T_g}(f_1, f_2, h)) \tag{7}$$

However, if we leave $H_f$ unconstrained, the maximum term can be arbitrarily large. In order to obtain a tight bound, we need to restrict the size of hypothesis space as well as maintain the upper bound. For $f_S \in H_1 \leq H_f$ and $f_T \in H_2 \leq H_f$, the following holds:

$$C_{S_g, T_g}(f_S, f_T, h) \leq \max_{f_1 \in H_1, f_2 \in H_2} C_{S_g, T_g}(f_1, f_2, h) \leq \max_{f_1, f_2 \in H_f} C_{S_g, T_g}(f_1, f_2, h) \tag{8}$$

The constrained subspace for $H_1$ is trivial as according to its definition, $f_S$ must belong to the space consisting of all classifiers for the source domain, namely $H_{sc}$. This implies $H_1 = H_{sc}$. However, the constrained subspace for $H_2$ is a little problematic since we have no access to the true labels of the target domain, thus it is hard to locate $f_T$. Therefore, we construct a hypothesis space for $H_2$ that most likely contains $f_T$.

As illustrated in Fig. 1c, if the conditional distributions are well-aligned (e.g., left side of Fig. 1c) after the matching between the source and target domain, it is fair to assume that $f_T \in H_{sc}$. However, the previous assumption of a good alignment might not always hold true, especially when samples from different classes are mixed together (e.g., right side of Fig. 1c). Considering this, we propose two training objectives in the following sections based on different constraints for $H_2$ which aim to alleviate the worst case caused by the unknown location of $f_T$.

### 3.2.1 Source-driven Hypothesis Space (SHS)

We assume $H_2$ is a space where the hypothesis can classify the samples from the source domain with an accuracy of $\gamma \in [0, 1]$, namely $H_{sc}^{\gamma}$, such that we can avoid the worst case by choosing a small value for the hyper-parameter $\gamma$ when a huge domain gap exists. In practice, it is difficult to actually build such a space and sample from it due to a huge computational cost. Instead, we use a weighted source error to constrain the behavior of $f_2$ as an approximation to the sample from $H_{sc}^{\gamma}$ (a weighted source error does not give us a classifier with an exact accuracy of $\gamma$; however, a small $\gamma$ allows $f_2$ to misclassify more source samples because the shared feature extractor $g$ will pay more attention on reducing the error on $f_1$), which leads to the final training objective (we omit some subscripts of the minimum and maximum term for simplicity):

$$\begin{cases} \min_{g,h}(\epsilon_{S_g}(h) + \max_{f_1, f_2} C_{S_g, T_g}(f_1, f_2, h)) \\ \min_{g, f_1, f_2}(\epsilon_{S_g}(f_1) + \gamma \epsilon_{S_g}(f_2)) \end{cases} \tag{9}$$

### 3.2.2 Target-driven Hypothesis Space (THS)

Firstly, we build a space consisting of all classifiers for the approximated target domain $\{(x_t^i, h(x_t^i)) \in X \times Y\}_{i=1}^m$ based on pseudo labels which can be obtained by predictions of $h$ during the training procedure, namely $H_{\tilde{t}c}$. Here, we assume $H_2$ is an intersection between two hypothesis spaces , i.e., $H_{sc}^{\eta} \cap H_{\tilde{t}c}^{1-\eta}$ where the hypothesis can classify samples from the source domain with an accuracy of $\eta \in [0, 1]$ and classify samples from the approximated target domain with an accuracy of $1 - \eta$. Analogously, the training objective is given by:

$$\begin{cases} \min_{g,h}(\epsilon_{S_g}(h) + \max_{f_1, f_2} C_{S_g, T_g}(f_1, f_2, h)) \\ \min_{g, f_1, f_2}(\epsilon_{S_g}(f_1) + \eta \epsilon_{S_g}(f_2) + (1 - \eta)\tilde{\epsilon}_{T_g}(f_2)) \end{cases} \tag{10}$$

### 3.2.3 Intuition to the Difference between Hypothesis Spaces

The reason we put the above constraints on hypothesis space $H_2$ can be intuitively explained by Fig. 2. If we confine the hypothesis space of $H_2$ to $H_{sc}$, the classifier is forced to correctly classify source samples. As a result, it increases the chance that all target samples are outside $f_2$. In this case, the feature extractor could move the samples into either side of the decision boundary to reduce the objective (shadow area in

Fig. 2a). By alleviating the hypothesis space (Fig. 2b illustrates THS and the same goes for SHS), we also alleviate restrictions on $f_2$ which makes it more likely that the decision boundary passes at least through some target samples, where the feature extractor can properly reduce the objective (striped area) by moving those samples (orange) toward the inside of $f_2$.

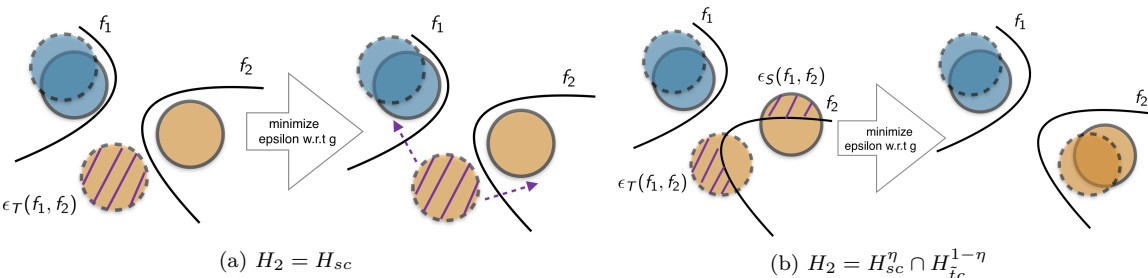

(a) $H_2 = H_{sc}$ (b) $H_2 = H_{sc}^{\eta} \cap H_{\tilde{t}c}^{1-\eta}$

Figure 2: (a) Improper constraint; (b) Alleviated constraint

## 3.3 Comparisons with Other Methods

In this section, we show that under some conditions, the proposed joint error based upper bound can reduce to other popular upper bounds, which demonstrates the generality of our proposal.

### 3.3.1 Margin Disparity Discrepancy

Zhang et al. (2019b) propose a novel margin-aware generalization bound based on scoring functions and define a new divergence MDD. The training objective used in MDD can be alternatively expressed as (here $\epsilon(h, f)$ denotes the margin disparity):

$$\min_{g,h}(\epsilon_{S_g}(h) + \max_f(\epsilon_{T_g}(f,h) - \epsilon_{S_g}(f,h))) \tag{11}$$

Recall Eq.9, if we set $f_1 = f_2 = f$ and free the constraint of $f$ to any $f \in H$, our proposal degrades exactly to MDD. As discussed above, the assumption about identical optimal labeling function $f_S = f_T$ is dangerous, because we have no idea about the location of $f_T$ and a perfect alignment between the conditional distributions of source and target domain is not likely to happen in practice. Besides, an unconstrained hypothesis space for $f$ is not helpful to build a tight bound.

### 3.3.2 Maximum Classifier Discrepancy

Saito et al. (2017b) propose two task-specific classifiers $f_1, f_2$ that are used to separate the decision boundary on the source domain, such that the feature extractor is encouraged to produce features near the support of the source samples. The objective used in MCD can be alternatively expressed as:

$$\begin{cases} \min_g(\epsilon_{S_g}(f_1) + \max_{f_1,f_2}(\epsilon_{T_g}(f_1,f_2))) \\ \min_{g,f_1,f_2}(\epsilon_{S_g}(f_1) + \epsilon_{S_g}(f_2)) \end{cases} \tag{12}$$

Recall Eq.9 and Eq.10, if we set $\gamma = 1$ (or $\eta = 1$) and $h = f_1$, our proposal reduces to MCD. As proved in 3.1, the proposed upper bound is optimized when $h = f_S$. However, it no longer holds after the upper bound is relaxed by taking the supremum, i.e., setting $h = f_1$ does not necessarily minimize the objective. Besides, as discussed above, assuming $H_2 = H_{sc}$ lacks generality since $f_T$ can be far away from $f_S$ and does not necessarily classify all source samples, which means the assumption of MCD is not likely to be applicable to those cases where a huge domain gap exists.

### 3.4 Cross Margin Discrepancy (CMD)

In this section, we propose a novel discrepancy measurement for $\epsilon$ instead of those commonly used ones (e.g., logistic, hinge, $L_1$). Following the above notations, we consider a score function $s(x, y)$ for multi-class classification where the output indicates the confidence of the prediction on class $y$. Thus an induced labeling function named $l_s$ from $X \rightarrow Y$ is given by:

$$l_s : x \rightarrow \underset{y \in Y}{\arg \max} \, s(x, y) \tag{13}$$

The margin between data points and the classification surface plays a significant role in achieving strong generalization performance. In order to quantify $\epsilon$ into a differentiable measurement as a surrogate of 0-1 loss, we introduce the margin theory developed by Koltchinskii & Panchenko (2002), where the margin loss is interpreted as:

$$\mathbb{E}_{(x,y) \in D}[\max(0, 1 + \max_{y' \neq y} s(x, y') - s(x, y))] \tag{14}$$

We aim to utilize this concept to further improve the reliability of our proposed method by leveraging this margin loss to define a novel measurement of the discrepancy between two hypotheses $f_1, f_2$ (e.g., a multi-layer perceptron whose output layer is a softmax function) over a distribution $D$, namely cross margin discrepancy:

$$\epsilon_D(f_1, f_2) = \mathbb{E}_{x \in D}[cmd(f_1, f_2, x)] \tag{15}$$

Before further discussion, we firstly construct two distributions $D_{f_1}, D_{f_2}$ induced by $f_1, f_2$ respectively, where $D_{f_1} = \{(x, l_{f_1}(x)) | x \sim D\}$ and $D_{f_2} = \{(x, l_{f_2}(x)) | x \sim D\}$. Then we consider the case where two hypotheses $(f_1, f_2)$ disagree, i.e., $y_1 = l_{f_1}(x) \neq l_{f_2}(x) = y_2$, and the primitive loss is defined as:

$$\begin{aligned} cmd(f_1, f_2, x) &= \log f_1(x, y_1) - \log f_2(x, y_1) + \log f_2(x, y_2) - \log f_1(x, y_2) \\ &= \log f_1(x, y_1) - \log f_1(x, y_2) + \log f_2(x, y_2) - \log f_2(x, y_1) \end{aligned} \tag{16}$$

Then the cross margin discrepancy $\epsilon_D(f_1, f_2)$ can be viewed as:

$$\mathbb{E}_{(x,y) \in D_{f_2}}[\max_{y' \neq y} \log f_1(x, y') - \log f_1(x, y)] + \mathbb{E}_{(x,y) \in D_{f_1}}[\max_{y' \neq y} \log f_2(x, y') - \log f_2(x, y)] \tag{17}$$

which is a sum of the margin loss for $f_1$ on $D_{f_2}$ and the margin loss for $f_2$ on $D_{f_1}$, if we use the logarithm of softmax as the score function.

Thanks to the trick introduced by Goodfellow et al. (2014) to mitigate the burden of exploding or vanishing gradients, we optimize a dual form of Eq.16 instead (we refer to $\log f(x, y)$ as the positive part and $\log(1 - f(x, y))$ as the negative part):

$$cmd(f_1, f_2, x) = \log f_1(x, y_1) + \log(1 - f_1(x, y_2)) + \log f_2(x, y_2) + \log(1 - f_2(x, y_1)) \tag{18}$$

During the training procedure, the two hypotheses will eventually agree on some points $(l_{f_1}(x) = l_{f_2}(x) = y)$ such that we need to define a new form of discrepancy measurement. Analogously, the primitive loss and its dual form are given by:

$$cmd(f_1, f_2, x) = \log \max(f_1(x, y), f_2(x, y)) - \log \min(f_1(x, y), f_2(x, y)) \tag{19}$$

$$cmd(f_1, f_2, x) = \log \max(f_1(x, y), f_2(x, y)) + \log \max(1 - f_1(x, y), 1 - f_2(x, y)) \tag{20}$$

Another reason why we propose such a discrepancy measurement is that it helps alleviate the instability in adversarial learning. As illustrated in Fig. 3b, during the optimization of a minimax game, when two hypotheses try to maximize the discrepancy (striped area), if one moves too fast around the decision boundary such that the discrepancy is actually maximized w.r.t some samples, then these samples can be moved into either side to decrease the discrepancy by tuning the feature extractor. From Fig. 3a, it can be seen that CMD is flat for the points around the original, i.e., the gradient w.r.t those points near the decision boundary will be relatively small, which can help to prevent the above failure as each update of a hypothesis will be subtle during the training.

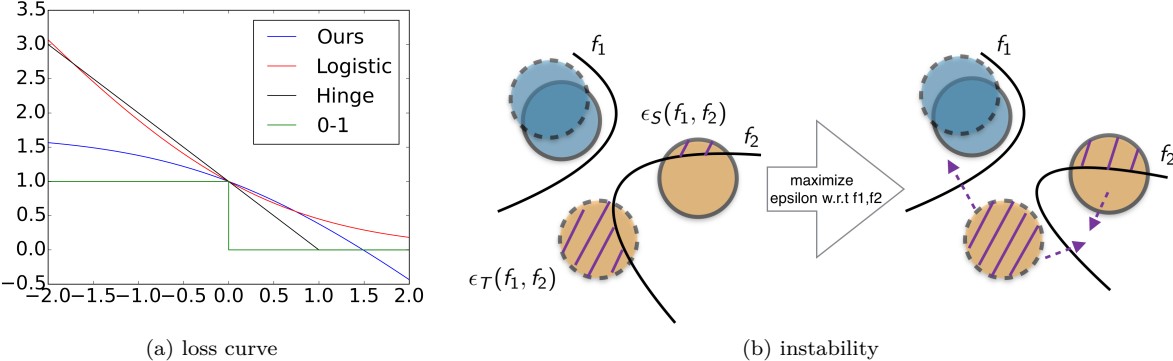

(a) loss curve          (b) instability

Figure 3: (a) Loss of binary classification where ours shows small gradient around the original; (b) Steep gradients near the original may lead to a extreme move of the decision boundary.

## 3.5 Training Procedure

This section briefly explains the training procedure of the proposed method, which follows a similar pipeline proposed by Saito et al. (2017b). Here we take SHS objective (Eq.9) as an example and each term is clearly defined below, where $g$ represents a feature extractor and $h, f_1, f_2$ are classifiers with a softmax output. First, we use the cross-entropy loss as a common surrogate of the source error:

$$\epsilon_{S_g}(h) = -\mathbb{E}_{x,y\sim S} \log(h(g(x), y)) \tag{21}$$

Then we define the discrepancy term with the proposed cross margin discrepancy (Eq.1, Eq.15, Eq.18, Eq.20):

$$C_{S_g,T_g}(f_1, f_2, h)) = \mathbb{E}_{x\sim T}[cmd(f_1, f_2, x) + cmd(h, f_1, x)] + \mathbb{E}_{x\sim S}[cmd(f_1, f_2, x) - cmd(h, f_2, x)] \tag{22}$$

Following the procedure described in Fig.4, for a mini-batch of input, during one training cycle, we firstly train $g, f_1, f_2$ to satisfy the hypothesis space constraint; secondly, we introduce a trade-off hyper-parameter $\lambda = 0.01$ to balance the source error and discrepancy then train $f_1, f_2$ to maximize the discrepancy term within the hypothesis space constraint; finally, we train $g, h$ to minimize the entire upper bound. We use $\hat{C}, \check{C}$ to emphasize the difference of the discrepancy term optimized in different steps such that the objective is consistent with GAN (the positive part of CMD is fixed in Step C). Step C is executed 4 times inside a training cycle according to Saito et al. (2017b). See A.5 for more details of the objective.

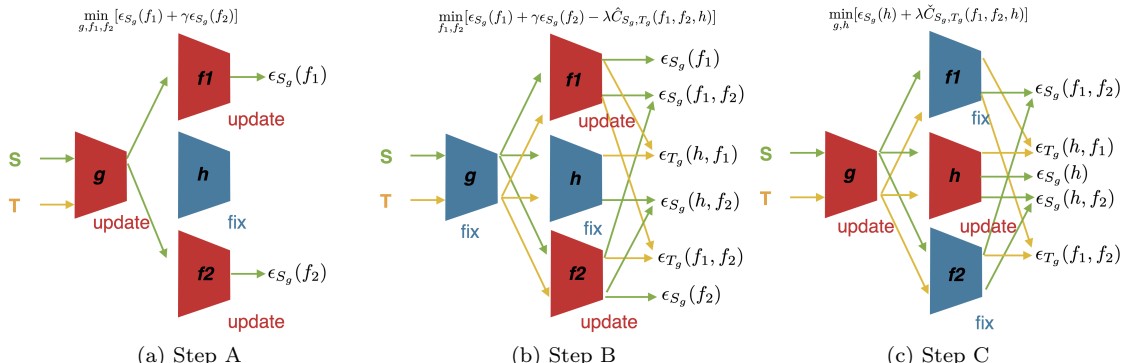

(a) Step A        (b) Step B        (c) Step C

Figure 4: Training procedure of the proposed method: (i) Train $g, f_1, f_2$ to satisfy the hypothesis space constraint; (ii) Train $f_1, f_2$ to maximize the discrepancy within the hypothesis space constraint; (iii) Train $g, h$ to minimize the entire upper bound.

Table 1: Accuracy of models adapted on digits datasets (* denotes the whole training set)

| METHOD | SVHN to MNIST | MNIST to SVHN | MNIST to USPS | MNIST* to USPS* | USPS to MNIST |
|---|---|---|---|---|---|
| Source Only | 67.1 | 21.3 | 76.7 | 79.7 | 63.4 |
| MDD[†](Long et al., 2015) | 71.1 | - | - | 81.1 | - |
| DANN[†]Ganin et al. (2016) | 71.1 | 25.1 | 77.3 | 85.1 | 73.2 |
| DRCN(Ghifary et al., 2016) | 82.0±0.1 | 40.1±0.1 | 91.8±0.1 | - | 73.7±0.1 |
| ADDA(Tzeng et al., 2017) | 76.0±1.8 | - | 89.4±0.2 | - | 90.1±0.8 |
| MCD(Saito et al., 2017b) | 96.2±0.4 | 11.2±1.1 | 94.2±0.7 | 96.5±0.3 | 94.1±0.3 |
| GPDA(Kim et al., 2019) | 98.2±0.1 | - | 96.5±0.2 | 98.1±0.1 | 96.4±0.1 |
| ours ($SHS + L_1, \gamma = 1$) | 96.8±0.2 | 30.4±1.5[4] | 94.5±0.3 | 96.8±0.3 | 95.2±0.2 |
| ours ($SHS + CMD, \gamma = 1$) | 97.5±0.2 | 31.5±1.8[4] | 95.3±0.3 | 97.2±0.2 | 95.6±0.2 |
| ours ($THS + CMD, \eta = 0$) | 98.6±0.1 | 50.3±1.3 | 96.8±0.2 | 97.9±0.1 | 96.9±0.1 |

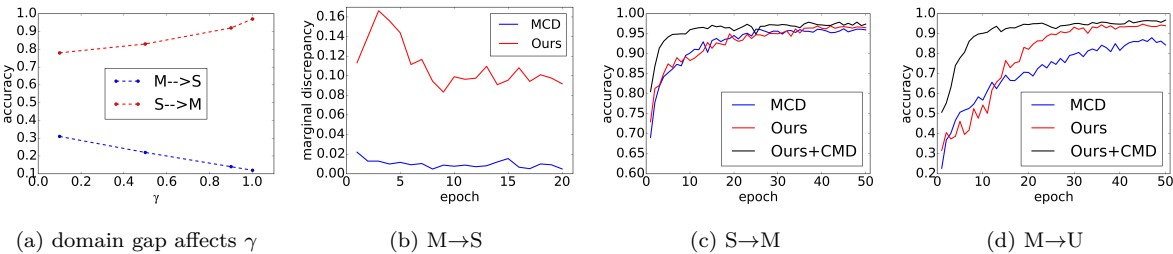

(a) domain gap affects $\gamma$     (b) M→S     (c) S→M     (d) M→U

Figure 5: (a) Sensitivity w.r.t. $\gamma$; (b) Comparisons for marginal discrepancy; (c)-(d) Comparisons for convergence rate; All plots are drawn based on the average of 3 runs.

## 4 Evaluation

In this section, we perform an extensive evaluation on image classification for several different datasets (Digit (Netzer et al., 2011; LeCun et al., 1998; Hull, 1994), VisDA (Peng et al., 2017), Office-Home (Venkateswara et al., 2017), Office-31 (Saenko et al., 2010)) to verify the effectiveness of our proposal. We choose MCD (Saito et al., 2017b) as a baseline for the major comparisons with ours because the two methods share the same concept, which is to tighten the upper bound established by Ben-David et al. (2010). Besides, the two methods utilize the same network architecture and training procedure, which makes the results fairly comparable. We conduct an ablation study (A.7) to show the contribution of each part of our proposal. We manually create an imbalance label situation and show the robustness of our proposal (A.6). Details of the experimental settings, the hyper-parameter selection and the training objective are provided in A.1, A.4, A.5.

### 4.1 Experiment on Digit Dataset

In this experiment, our proposal is assessed in four types of adaptation scenarios by adopting commonly used digits datasets, i.e., MNIST (LeCun et al., 1998), Street View House Numbers (SVHN) (Netzer et al., 2011), and USPS (Hull, 1994) such that the result can be easily compared with other popular methods. All experiments are performed in an unsupervised fashion without any kinds of data augmentation. We report the accuracy of different methods in Tab. 1. Our proposal manages to improve the performance in almost all settings except for a single result compared with GPDA (Kim et al., 2019). However, their solution requires sampling that increases data size and is equivalent to adding Gaussian noise to the last layer of a classifier, which is considered as a type of augmentation.

---

[4]We use $\gamma = 0.1$ for MNIST → SVHN

Table 2: Accuracy of ResNet-101 model fine-tuned on VisDA dataset within 10 epochs

| METHOD | plane | bcycl | bus | car | horse | knife | mcycl | person | plant | sktbrd | train | truck | avg |
|---|---|---|---|---|---|---|---|---|---|---|---|---|---|
| Source Only | 55.1 | 53.3 | 61.9 | 59.1 | 80.6 | 17.9 | 79.7 | 31.2 | 81.0 | 26.5 | 73.5 | 8.5 | 52.4 |
| MDD(Long et al., 2015) | 87.1 | 63.0 | 76.5 | 42.0 | 90.3 | 42.9 | 85.9 | 53.1 | 49.7 | 36.3 | 85.8 | 20.7 | 61.1 |
| DANN(Ganin et al., 2016) | 81.9 | 77.7 | 82.8 | 44.3 | 81.2 | 29.5 | 65.1 | 28.6 | 51.9 | 54.6 | 82.8 | 7.8 | 57.4 |
| MCD(Saito et al., 2017b) | 87.0 | 60.9 | 83.7 | 64.0 | 88.9 | 79.6 | 84.7 | 76.9 | 88.6 | 40.3 | 83.0 | 25.8 | 71.9 |
| GPDA(Kim et al., 2019) | 83.0 | 74.3 | 80.4 | 66.0 | 87.6 | 75.3 | 83.8 | 73.1 | 90.1 | 57.3 | 80.2 | 37.9 | 73.3 |
| MCCJin et al. (2020) | 88.1 | 80.3 | 80.5 | 71.5 | 90.1 | 93.2 | 85.0 | 71.6 | 89.4 | 73.8 | 85.0 | 36.9 | 78.8 |
| ours ($SHS + L_1, \gamma = 1$) | 86.3 | 82.7 | 83.7 | 68.7 | 87.9 | 72.7 | 85.4 | 61.5 | 87.3 | 55.5 | 75.2 | 34.1 | 73.4 |
| ours ($SHS + CMD, \gamma = 1$) | 88.4 | 83.3 | 74.8 | 78.0 | 88.1 | 43.2 | 88.2 | 68.9 | 87.6 | 65.5 | 92.6 | 58.5 | 76.4 |
| ours ($THS + CMD, \eta = 0.9$) | 93.8 | 79.5 | 79.3 | 55.9 | 93.9 | 93.8 | 86.5 | 80.3 | 91.6 | 87.7 | 85.4 | 51.6 | 81.6 |

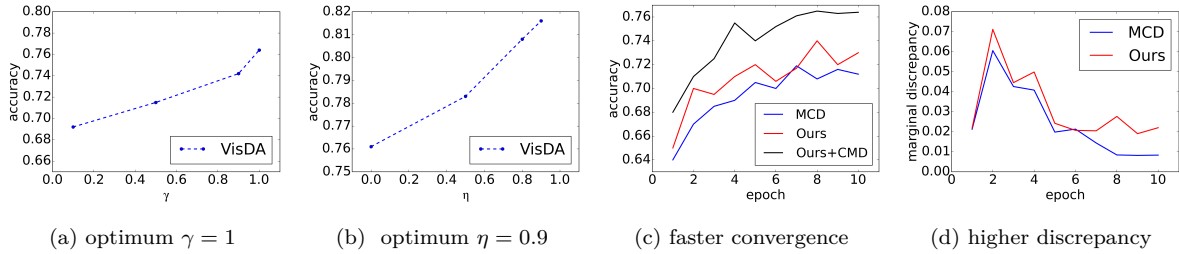

(a) optimum $\gamma = 1$      (b) optimum $\eta = 0.9$      (c) faster convergence      (d) higher discrepancy

Figure 6: (a)-(b) Sensitivity w.r.t. $\gamma,\eta$; (c) Comparisons for convergence rate; (d) Comparisons for marginal discrepancy; All plots are drawn based on the average of 3 runs.

Fig. 5a shows the result of SHS+CMD proposal and it can be seen that the hyper-parameter $\gamma$ is quite sensitive to the domain gap. In short, setting $\gamma = 1$ here yields the best performance in most situations, since $f_S, f_T$ can be quite close after aligning distributions, especially in these easily adapted scenarios. However, in MNIST $\rightarrow$ SVHN, setting $\gamma = 0.1$ gives the optimum which means that $f_S, f_T$ are so far away due to a huge domain gap that no extractor is capable of introducing an identical conditional distribution in feature space. Besides, for THS+CMD proposal, we test the performance on different value of $\eta$ then find out that the accuracy remains high and varies subtly, therefore we omit a figure for it. Furthermore, Fig. 5b empirically proves simply minimizing the discrepancy between marginal distributions does not necessarily lead to a reliable adaptation, which demonstrates the importance of the joint error (because MCD and SHS+L$_1$ are directly comparable). In addition, Fig. 5c and Fig. 5d show the effectiveness of the cross margin discrepancy which accelerates the convergence and provides a slightly better result.

## 4.2 Experiment on VisDA Dataset

The VisDA dataset (Peng et al., 2017) is designed for 12-class adaptation task from synthetic object to real object images. Source domain contains 152,397 synthetic images generated by rendering 3D CAD models and target domain is collected from MSCOCO (Lin et al., 2014) consisting of 55,388 real images. We report the accuracy of different methods in Tab. 2, and find that our proposal works well in all settings. The image structure of this dataset is more complex than that of digits, yet our method provides reliable performance even under such a challenging condition. Another key observation is that some competing methods (e.g., DANN, MCD), which can be categorized as distribution matching based on adversarial learning, perform worse than MDD (which simply matches statistics) in classes such as plane and horse, while our methods perform better across all classes. This clearly demonstrates the importance of the joint error. As for the SHS+CMD proposal (Fig. 6a), performance drops when relaxing the constraint which actually confuses us. Because we expect an improvement here since it is hard to believe that $f_S, f_T$ eventually lie in a similar space judging from the relatively low prediction accuracy. As for the THS+CMD proposal (Fig. 6b), we test the adaptation performance for different $\eta$ and the prediction accuracy drops when $\eta$ goes below 0.9. One possible cause is that $f_2$ and $h$ might almost agree on the target domain, such that the prediction of $h$ cannot provide more accurate information of the target domain without introducing noisy pseudo labels. Fig. 6c and

Table 3: The accuracy of ResNet-50 model fine-tuned on the Office-Home dataset. We repeated each experiment 5 times and report the average of the accuracy.

| METHOD | Ar→Cl | Ar→Pr | Ar→Rw | Cl→Ar | Cl→Pr | Cl→Rw | Pr→Ar | Pr→Cl | Pr→Rw | Rw→Ar | Rw→Cl | Rw→Pr | Avg |
|---|---|---|---|---|---|---|---|---|---|---|---|---|---|
| Source Only | 34.9 | 50.0 | 58.0 | 37.4 | 41.9 | 46.2 | 38.5 | 31.2 | 60.4 | 53.9 | 41.2 | 59.9 | 46.1 |
| DANN(Ganin et al., 2016) | 45.6 | 59.3 | 70.1 | 47.0 | 58.5 | 60.9 | 46.1 | 43.7 | 68.5 | 63.2 | 51.8 | 76.8 | 57.6 |
| MCD(Saito et al., 2017b) | 51.9 | 70.7 | 74.8 | 59.0 | 68.4 | 68.8 | 58.2 | 51.6 | 75.1 | 69.5 | 55.8 | 79.3 | 65.3 |
| CDAN(Long et al., 2018) | 50.7 | 70.6 | 76.0 | 57.6 | 70.0 | 70.0 | 57.4 | 50.9 | 77.3 | 70.9 | 56.7 | 81.6 | 65.8 |
| SymNets(Zhang et al., 2019a) | 47.7 | 72.9 | 78.5 | 64.2 | 71.3 | 74.2 | 64.2 | 48.8 | 79.5 | 74.5 | 52.6 | 82.7 | 67.6 |
| SPL(Wang & Breckon, 2020) | 54.5 | 77.8 | 81.9 | 65.1 | 78.0 | 81.1 | 66.0 | 53.1 | 82.8 | 69.9 | 55.3 | 86.0 | 71.0 |
| AADA(Yang et al., 2020) | 54.0 | 71.3 | 77.5 | 60.8 | 70.8 | 71.2 | 59.1 | 51.8 | 76.9 | 71.0 | 57.4 | 81.8 | 67.0 |
| SRDC(Tang et al., 2020) | 52.3 | 76.3 | 81.0 | 69.5 | 76.2 | 78.0 | 68.7 | 53.8 | 81.7 | 76.3 | 57.1 | 85.0 | 71.3 |
| SCAL(Wang et al., 2022) | 55.3 | 72.7 | 78.7 | 63.1 | 71.7 | 73.5 | 61.4 | 51.6 | 79.9 | 72.5 | 57.8 | 81.0 | 68.3 |
| ours $(THS + CMD, \eta = 0.9)$ | 60.3 | 77.8 | 81.0 | 66.0 | 74.4 | 74.5 | 66.7 | 59.3 | 81.8 | 74.2 | 62.7 | 84.9 | 72.0 |

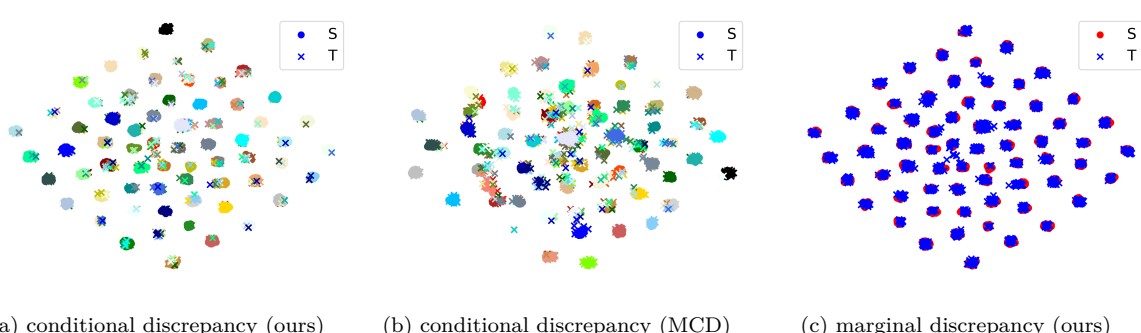

(a) conditional discrepancy (ours)  (b) conditional discrepancy (MCD)  (c) marginal discrepancy (ours)

Figure 7: Feature visualization with t-SNE in scenario Real to Art; (a)-(b) shows the conditional discrepancy by using different colors for each class; (c) shows the marginal discrepancy by using the same color for the all classes.

Fig. 6d again demonstrate the effectiveness of the cross margin discrepancy and the importance of the joint error.

## 4.3 Experiment on Office-Home Dataset

Office-Home (Venkateswara et al., 2017) is a complex dataset containing 15,500 images from four significantly different domains: Art (Ar), Clipart (Cl), Product (Pr) and Real-World (Rw). We report the accuracy of different methods in Tab. 3. From the table, it can be seen that the adaptation accuracy of the source-only method is rather low, which means a huge domain gap is quite likely to exist. In such case, we believe simply minimizing the discrepancy between the source and target domain may not work as the joint error can increase when aligning the distributions. As reported, our proposal gives better performance in general when the adaptation is hard (1st, 8th and 11th columns). This result suggests that our proposal which incorporates the joint error into the target error upper bound can boost the performance especially when there is a large domain gap. SPL (Wang & Breckon, 2020) and SRDC (Tang et al., 2020) share a similar strategy which can be described as cluster based iterative pseudo labeling. Despite the outstanding performance, their methods lack a theoretical guarantee on the generalization error bound. Besides, they assume samples in the target domain are well clustered within the feature space and can be labeled by the euclidean distance from clusters obtained with K-means. As for the first assumption, it is not always true since the high dimensional features can lie in a low dimensional manifold (e.g., Swiss Roll dataset) where no cluster exists. As for the second assumption, the labeling scheme can be vulnerable because K-means is sensitive to the initialization and the euclidean distance can suffer from the curse of dimensionality where there is little difference in the distances between different pairs of samples. After all, this works aims to demonstrate the necessity of joint error especially when the domain gap is huge; thus further discussion on the pseudo labeling strategies is beyond the scope of this work.

In addition, we plot the learned features with t-SNE (van der Maaten & Hinton, 2008) in Fig. 7 which shows the scenario of Real to Art . In Fig. 7a, we illustrate the source (dot) and target (cross) features with

Table 4: The accuracy of ResNet-50 model fine-tuned on the Office-31 dataset. We repeated each experiment 5 times and report the average and the standard deviation of the accuracy.

| METHOD | A→W | D→W | W→D | A→D | D→A | W→A | Avg |
|---|---|---|---|---|---|---|---|
| Source Only | 68.4±0.2 | 96.7±0.1 | 99.3±0.1 | 68.9±0.2 | 62.5±0.3 | 60.7±0.3 | 76.1 |
| DANN(Ganin et al., 2016) | 82.0±0.4 | 96.9±0.2 | 99.1±0.1 | 79.7±0.4 | 68.2±0.4 | 67.4±0.5 | 82.2 |
| ADDA(Tzeng et al., 2017) | 86.2±0.5 | 96.2±0.3 | 98.4±0.3 | 77.8±0.3 | 69.5±0.4 | 68.9±0.5 | 82.9 |
| MCD(Saito et al., 2017b) | 88.6±0.2 | 98.5±0.1 | 100.0±.0 | 92.2±0.2 | 69.5±0.1 | 69.7±0.3 | 86.5 |
| CDAN(Long et al., 2018) | 94.1±0.1 | 98.6±0.1 | 100.0±.0 | 92.9±0.2 | 71.0±0.3 | 69.3±0.3 | 87.7 |
| SymNets(Zhang et al., 2019a) | 90.8±0.1 | 98.8±0.3 | 100.0±0.0 | 93.9±0.5 | 74.6±0.6 | 72.5±0.5 | 88.4 |
| SPL(Wang & Breckon, 2020) | 92.7±0.0 | 98.1±0.0 | 99.8±0.0 | 93.7±0.0 | 76.4±0.0 | 76.9±0.0 | 89.6 |
| MCC(Jin et al., 2020) | 95.5±0.2 | 98.6±0.1 | 100.0±0.0 | 94.4±0.3 | 72.9±0.2 | 74.9±0.3 | 89.4 |
| SRDC(Tang et al., 2020) | 94.6±1.0 | 99.2±0.5 | 100.0±0.0 | 92.6±0.6 | 78.1±1.3 | 76.3±0.2 | 90.1 |
| SCAL(Wang et al., 2022) | 93.5±0.2 | 98.5±0.1 | 100.0±0.0 | 93.4±0.3 | 72.4±0.1 | 74.0±0.3 | 88.6 |
| ours $(THS + CMD, \eta = 0.9)$ | 91.9±0.5 | 99.0±0.2 | 100.0±.0 | 93.7±0.5 | 76.1±0.2 | 77.8±0.2 | 89.8 |

different colors that represent their classes. In our method, most of the target features are clustered to their corresponding sources and do not show a large variance from the class center compared to MCD (Fig. 7b). We also visualize features of the source (red dot) and target (blue cross) domain in Fig. 7c to show the entire domain discrepancy. It can be seen that the target features are also well-aligned with those of the source domain.

### 4.4 Experiment on Office-31 Dataset

Office-31 (Saenko et al., 2010) is a popular dataset to verify the effectiveness of a domain adaptation algorithm, which contains three diverse domains: Amazon (A), Webcam (W) and DSLR (D) with 4,652 images in 31 unbalanced classes. The results on Office-31 are reported in Tab. 4. As for the tasks D→A and W→A, judging from the relatively low adaptation accuracy across all methods, it is quite likely that there is a huge domain gap between the source and target domain. Our method works well in such cases which demonstrates that the proposal manages to penalize the undesired matching between the source and target domain. However, the advantage is not remarkable especially for the task A→W, where our proposal shows relatively high variance and poor performance. One possible reason is that our method depends on building reliable classifiers for the source domain to satisfy the constraint for the hypothesis space. However, the Amazon domain contains a lot of noise and the entire dataset lacks diversity (e.g., some domains only have several hundred samples with many duplicates) which makes it difficult for a traditional error bound based method that usually requires a sufficient sample size to learn a reliable classifier.

## 5 Conclusion

In this work, we propose a novel upper bound that takes the joint error into account. Then we further pursue a tighter bound with reasonable constraints on the hypothesis space. Additionally, we adopt a novel cross domain discrepancy for dissimilarity measurement which alleviates the instability during adversarial learning. Extensive empirical evidence shows that an invariant representation is not enough to guarantee a good generalization performance in the target domain, as the joint error matters especially when the domain gap is huge. We believe our results take an important step towards understanding unsupervised domain adaptation, and also stimulate future work on the design of stronger adaptation algorithms that manage to align conditional distributions without using pseudo-labels from the target domain.

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

## A  Appendix

### A.1  Experimental Setting

### A.1.1  Digits Dataset

**SVHN ↔ MNIST** We firstly examine the adaptation from SVHN (Fig. 8a) to MNIST (Fig. 8b). We use the standard training set and testing set for both source and target domains. The feature extractor contains

three $5 \times 5$ convolutional layers with stride two $3 \times 3$ max pooling placed after the first two convolutional layers and a single fully-connected layer. For classifiers, we use 2-layer fully-connected networks.

**MNIST $\leftrightarrow$ USPS** As for the adaptation between MNIST and USPS (Fig. 8c), we follow the training protocol established in Long et al. (2013) by sampling 2000 images from MNIST and 1800 from USPS. As for the test samples , the standard version is used for both source and target domains. The feature generator contains two $5 \times 5$ convolutional layers with stride two $2 \times 2$ max pooling placed after each convolutional layer and a single fully-connected layer. For classifiers, we use 2-layer fully-connected networks.

Throughout the experiment, we employ the CNN architecture used in Saito et al. (2017b), where batch normalization is applied to each layer and a 0.5 rate of dropout is used between fully connected layers. Besides, spectral normalization (Miyato et al., 2018) is deployed for the classifiers in all of the following experiments to stabilize adversarial learning. The major reason we utilize this technique is because we observe a performance drop when the network is over-trained (it usually occurs after several hundreds epochs). We are still not sure whether it is caused by a over-fitting on the source error or a general neural network training problem related to early stop (this phenomenon can also be observed in other algorithms). One common solution is to use a gradient reversal layer(Ganin et al., 2016) to balance the weight between the source error and the discrepancy between the two domains. However, it will introduce more hyper-parameters to tune. Besides, in our proposed cross margin discrepancy, as is illustrated in Fig. 9, we maximize $\log f_1(x, y_1) + \log(1 - f_2(x, y_1))$ w.r.t the classifiers but minimize only $\log(1 - f_2(x, y_1))$ (a part of the objective) w.r.t the feature extractor to avoid unnecessary oscillation during adversarial learning. This implies the objective we try to maximize is not equivalent to that we try to minimize. And in such case, it is difficult to employ gradient reversal layer which is supposed to train the classifiers and the feature extractor in a single objective simultaneously. With the help of spectral normalization, we can ensure the classifiers is approximately Lipschitz such that the performance drop at an early stage can be avoided since the gradient w.r.t the classifiers will be relatively small. Adam (Kingma & Ba, 2014) is used for optimization with a mini-batch size of 128 and a learning rate of $10^{-4}$. As for the SHS proposal, we verify the sensitivity of our model on various value for $\gamma = \{0.1, 0.5, 0.9, 1.0\}$. As for the THS proposal, setting $\eta = 0$ usually provides a reliable performance and the adaptation result changes subtly for different $\eta$.

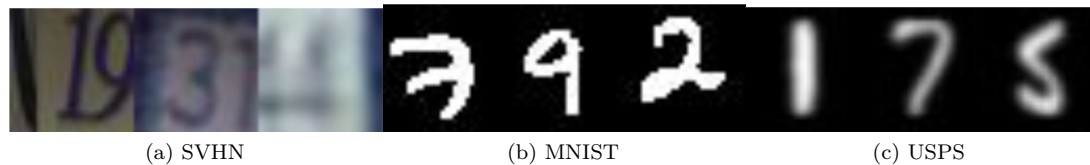

| (a) SVHN | (b) MNIST | (c) USPS |

Figure 8: Random samples from each dataset.

### A.1.2 VisDA Dataset

We further evaluate our method on object classification. The VisDA dataset (Peng et al., 2017) is used here, which is designed for 12-class adaptation task from synthetic object to real object images. Source domain contains 152,397 synthetic images, which are generated by rendering 3D CAD models. Data of the target domain is collected from MSCOCO (Lin et al., 2014) consisting of 55,388 real images. Since the 3D models are generated without the background and color diversity, the synthetic domain is quite different from the real domain, which makes it a much more difficult problem than digits adaptation. Again, this experiment is performed in unsupervised fashion and no data augmentation technique excluding horizontal flipping is allowed.

Following the protocol established in Saito et al. (2017b), we evaluate our method by fine-tuning a ResNet-101 (He et al., 2015) model pretrained on ImageNet (Deng et al., 2009). The model except the last layer combined with a single-layer bottleneck is used as feature extractor and a randomly initialized 2-layer fully-connected network is used as a classifier, where batch normalization is applied to each layer and a 0.5 rate of dropout is conducted. Nesterov accelerated gradient is used for optimization with a mini-batch size of 32 and an initial learning rate of $10^{-3}$ which decays exponentially. The network architecture used in Saito et al. (2017b) is

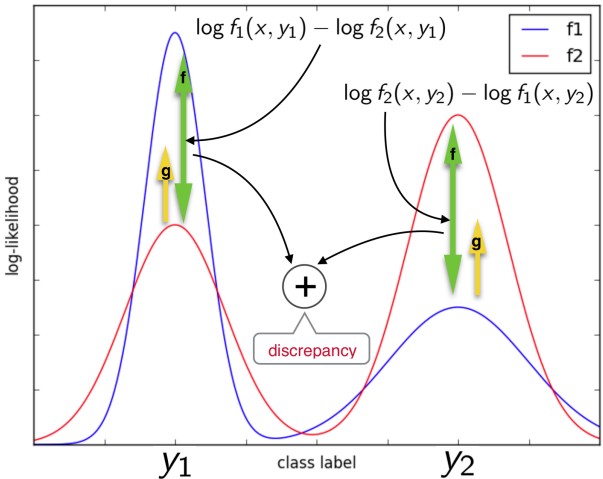

Figure 9: Illustrated details for the cross margin discrepancy

originally a 3-layer fully-connected classifier. However, as is mentioned in the main paper, we show that a relatively weaker classifier will improve the stability of the adversarial learning. Therefore, we take advantage of the framework proposed in Zhang et al. (2019b) by regarding the top layer of the classifier, i.e., a bottleneck, as a part of the feature extractor (it does not change the entire complexity of the model). Another crucial point here is that the network parameters w.r.t. the ResNet-101 and the bottleneck are updated separately at a ratio of 1:10, which is the same with Zhang et al. (2019b). A similar trick is used in Saito et al. (2017b) to relatively slow down the updating for ResNet-101 by introducing a Nesterov optimizer with a momentum term for the classifier. The common reason here that we do not want the feature extractor to be updated too much is very simple as ResNet-101 is so powerful that it can easily over-fit the source risk. We apply horizontal flipping of input images during training as the only data augmentation. As for the hyper-parameter, we test for $\gamma = \{0.1, 0.5, 0.9, 1\}$ and $\eta = \{0, 0.5, 0.8, 0.9\}$. For a direct comparison, we report the accuracy after 10 epochs.

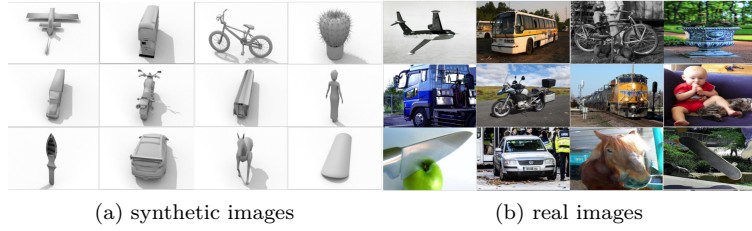

(a) synthetic images          (b) real images

Figure 10: (a) Samples from source domain. (b) Samples from target domain.

### A.1.3 Office-Home Dataset

Office-Home (Venkateswara et al., 2017) is a complex dataset (Fig. 11) containing 15,500 images from four significantly different domains: Art (paintings, sketches and/or artistic depictions), Clipart (clip art images), Product (images without background), and Real-world (regular images captured with a camera). In this experiment, following the protocol from Zhang et al. (2019b), we evaluate our method by fine-tuning a ResNet-50 (He et al., 2015) model pretrained on ImageNet (Deng et al., 2009). The model except the last layer combined with a single-layer bottleneck is used as feature extractor and a randomly initialized 2-layer fully-connected network with width 1024 is used as a classifier, where batch normalization is applied to each layer and a 0.5 rate of dropout is conducted. For optimization, we use the SGD with the Nesterov momentum term fixed to 0.9, where the batch size is 32 and learning rate is adjusted according to Ganin et al. (2016).We

apply horizontal flipping and resized cropping of input images during training as the data augmentation similar to Zhang et al. (2019b).

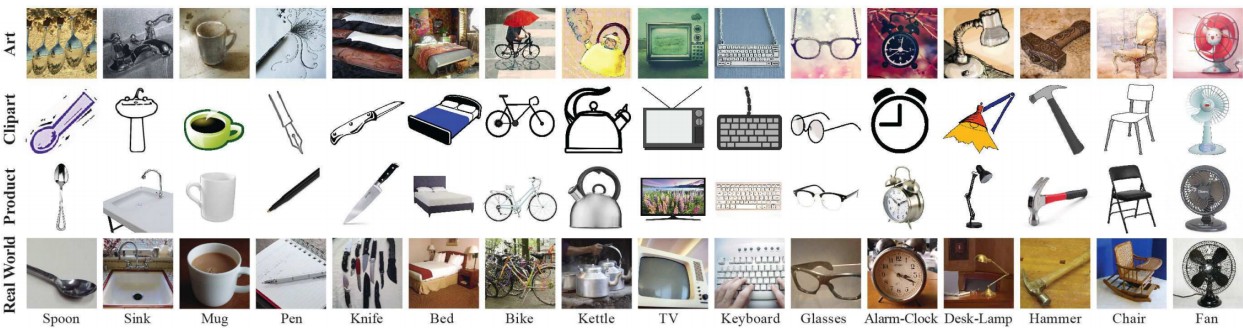

Figure 11: Sample images from the Office-Home dataset (Venkateswara et al., 2017).

### A.1.4 Office-31 Dataset

Office-31 (Saenko et al., 2010) (Fig. 12) is a popular dataset to verify the effectiveness of a domain adaptation algorithm, which contains three diverse domains, Amazon from Amazon website, Webcam by web camera and DSLR by digital SLR camera with 4,652 images in 31 unbalanced classes. Besides, we show several noisy samples in Fig.13 to backup our assumption for the failure case. In this experiment, following the protocol from Zhang et al. (2019b), we evaluate our method by fine-tuning a ResNet-50 (He et al., 2015) model pretrained on ImageNet (Deng et al., 2009). The model used here is almost identical to the one in Office-Home experiment except a different width 2048 for classifiers. For optimization, we use the SGD with the Nesterov momentum term fixed to 0.9, where the batch size is 32 and learning rate is adjusted according to Ganin et al. (2016). We apply horizontal flipping and resized cropping of input images during training as the data augmentation similar to Zhang et al. (2019b).

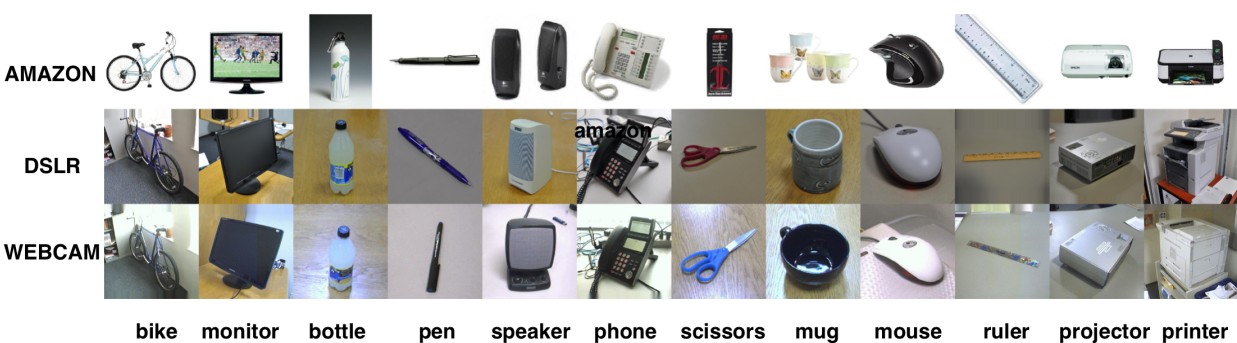

Figure 12: Sample images from the Office-31 dataset.

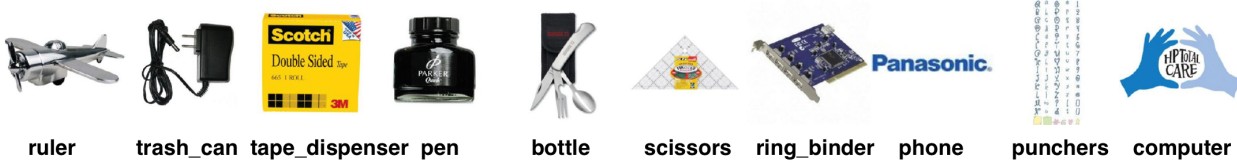

Figure 13: Noisy images from the Office-31 dataset.

## A.2 Rademacher Complexity

Let $H$ be a set of real-valued functions defined over a set $X$. Given a sample $S \in X^m$, the empirical Rademacher complexity of $H$ is defined as follows:

$$\hat{\Re}_S(H) = \frac{2}{m} \mathbb{E}_\sigma \left[ \sup_{h \in H} |\sigma_i h(x_i)| \,\middle|\, S = (x_1, ..., x_m) \right] \tag{23}$$

The expectation is taken over $\sigma = (\sigma_1, ..., \sigma_n)$ where $\sigma_i$ is an independent uniform random variable taking values in $\{-1, +1\}$.

The Rademacher complexity measures the ability of a class of functions to fit noise and also has the additional advantage that it is data-dependent and can be measured from finite samples, which can lead to tighter bounds than those based on other measures of complexity such as the VC-dimension. Following the established theory proposed by Mansour et al. (2009), we denote the empirical average of a hypothesis $h : X \to \{0,1\}$ by $\hat{R}(h)$ and its expectation over samples drawn according to the distribution considered by $R(h)$. The following gives a version of the Rademacher complexity bounds (Koltchinskii & Panchenko, 2000; Bartlett & Mendelson, 2002):

Let $H$ be a class of functions mapping $X \times Y \to [0,1]$ and $S = ((x_1, y_1), ..., (x_m, y_m))$ a finite sample drawn i.i.d. according to a distribution $Q$. Then, for any $\sigma > 0$, with probability at least $1 - \sigma$ over samples $S$ of size $m$, the following inequality holds for all $h \in H$:

$$R(h) \leq \hat{R}(h) + \hat{\Re}_S(H) + 3\sqrt{\frac{\log \frac{2}{\sigma}}{2m}} \tag{24}$$

Recall our proposed upper bound, we have: $\epsilon_T(h) \leq \epsilon_S(h) + C_{S,T}(f_S, f_T, h)$ where $C_{S,T}$ is further bounded by: $\sup_{f_1 \in H_1, f_2 \in H_2} \{\epsilon_T(f_1, f_2) + \epsilon_S(f_1, f_2) + \epsilon_T(h, f_1) - \epsilon_S(h, f_2)\}$. Now we relax this bound by applying $\epsilon_S(f_1, f_2) \leq \epsilon_S(f_1, h) + \epsilon_S(f_2, h)$ and name it $d(S, T; h) = \sup_{f_1 \in H_1, f_2 \in H_2} \{\epsilon_T(f_1, f_2) + \epsilon_S(f_1, h) + \epsilon_T(f_1, h)\}$. Assume that the loss function $\epsilon$ is bounded by $M > 0$:$\epsilon(h, h') \leq M$ for all $h, h' \in H$. Let $Q$ be a distribution over $X$ and let $\hat{Q}$ denote the corresponding empirical distribution for a sample $S = (x_1, ..., x_m)$. Then we can scale the loss $\epsilon$ to $[0, 1]$ by dividing by $M$, and denote the new class by $\epsilon_{H_1, H_2}$ which represents the class of functions $\{x \to \epsilon(f_1(x), f_2(x)) : f_1 \in H_1, f_2 \in H_2\}$. By the above theorem, for any $\sigma > 0$, with probability at least $1 - \sigma$, the following inequality holds for all $f_1 \in H_1, f_2 \in H_2$:

$$\frac{\epsilon_Q(f_1, f_2)}{M} \leq \frac{\epsilon_{\hat{Q}}(f_1, f_2)}{M} + \hat{\Re}_S(\epsilon_{H_1, H_2}/M) + 3\sqrt{\frac{\log \frac{2}{\sigma}}{2m}} \tag{25}$$

The empirical Rademacher complexity has the property that $\hat{\Re}(\alpha H) = \alpha \hat{\Re}(H)$ for any hypothesis class $H$ and positive number $\alpha$, which helps simplify the above bound. Now let $S$ be a distribution over $X$ and $\hat{S}$ the corresponding empirical distribution for a sample $\tilde{S}$, and let $T$ be a distribution over $X$ and $\hat{T}$ the corresponding empirical distribution for a sample $\tilde{T}$. Then, for any $\sigma > 0$, with probability at least $1 - \sigma$ over samples $\tilde{S}$ of size $m$ drawn according to $S$ and samples $\tilde{T}$ of size $n$ drawn according to $T$, we can write:

$$\begin{aligned} d(S, T; h) \leq &\sup_{f_1 \in H_1, f_2 \in H_2} \{\epsilon_{\hat{T}}(f_1, f_2) + \epsilon_{\hat{S}}(f_1, h) + \epsilon_{\hat{T}}(f_1, h)\} \\ &+ \hat{\Re}_{\tilde{T}}(\epsilon_{H_1, H_2}) + \hat{\Re}_{\tilde{S}}(\epsilon_{H_1}) + \hat{\Re}_{\tilde{T}}(\epsilon_{H_1}) \\ &+ 3M\left(\sqrt{\frac{\log \frac{2}{\sigma}}{2m}} + 2\sqrt{\frac{\log \frac{2}{\sigma}}{2n}}\right) \end{aligned} \tag{26}$$

## A.3 Compatibility

We do agree that it is not easy to find the true labeling function inside a specific hypothesis space. However, we believe it is a feasible assumption that labeling functions lie in a hypothesis space with enough complexity.

For instance, Mansour et al. (2009) makes a similar assumption that H includes $f_T$. Besides, since the algorithm is always run within finite samples, there is quite likely to be a function inside a specific hypothesis space that could perfectly mimic the behavior of the true labeling function on those samples. Therefore, even if the hypothesis space we use does not contain the true labeling function, it will not harm the actual learning process. The proof is as below:

$$
\begin{aligned}
\epsilon_T(h) &= \epsilon_T(h, f_T) \\
&= \epsilon_T(h, f_T) - \epsilon_T(h, f_S) + \epsilon_T(h, f_S) + \epsilon_S(h, f_S) - \epsilon_S(h, f_S) + \epsilon_S(h, f_T) - \epsilon_S(h, f_T) \\
&= \epsilon_S(h, f_S) + (\epsilon_T(h, f_T) - \epsilon_T(h, f_S)) + (\epsilon_S(h, f_T) - \epsilon_S(h, f_S)) + \epsilon_T(h, f_S) - \epsilon_S(h, f_T) \\
&\leq \epsilon_S(h) + \epsilon_T(f_S, f_T) + \epsilon_S(f_S, f_T) + \epsilon_T(h, f_S) - \epsilon_S(h, f_T) \\
&\leq \epsilon_S(h) + \epsilon_T(f_S, f_S^*) + \epsilon_T(f_S^*, f_T^*) + \epsilon_T(f_T^*, f_T) + \epsilon_S(f_S, f_S^*) + \epsilon_S(f_S^*, f_T^*) + \epsilon_S(f_T^*, f_T) \\
&\quad + \epsilon_T(h, f_S^*) + \epsilon_T(f_S^*, f_S) - \epsilon_S(h, f_T^*) + \epsilon_S(f_T^*, f_T) \\
&= \epsilon_S(h) + C_{S,T}(f_S^*, f_T^*, h) + \theta
\end{aligned}
\tag{27}
$$

where $\theta = 2\epsilon_T(f_S, f_S^*) + \epsilon_S(f_S, f_S^*) + 2\epsilon_S(f_T^*, f_T) + \epsilon_T(f_T^*, f_T)$.

We assume $f_S^* = \arg\min_{f \in H} \epsilon_{S \bigcup 2T}(f_S, f)$ and $f_T^* = \arg\min_{f \in H} \epsilon_{2S \bigcup T}(f_T, f)$. If $f_S^*, f_T^* \in H$ can perfectly mimic the behavior of $f_S, f_T$ on $S$ and $T$, then $\theta$ can be totally ignored and the obtained bound become compatible with the original upper bound by replacing $f_S, f_T$ with $f_S^*, f_T^*$.

## A.4    Hyperparameter Selection

In all experiments, the hyper-parameters $\gamma, \eta$ are set based on the validation performance on a separate dataset consisting of a few hundreds of labeled target samples. As showed in the main paper, it can be seen that the performance of our proposal is quite sensitive to the change of the hyper-parameters. However, it also suggests that an appropriate choice of the hyper-parameters can significantly boost the performance, which allows us to select the hyper-parameters via a validation dataset. Fig. 14 shows two examples of the validation performance on different adaptation tasks w.r.t. $\gamma, \eta$.

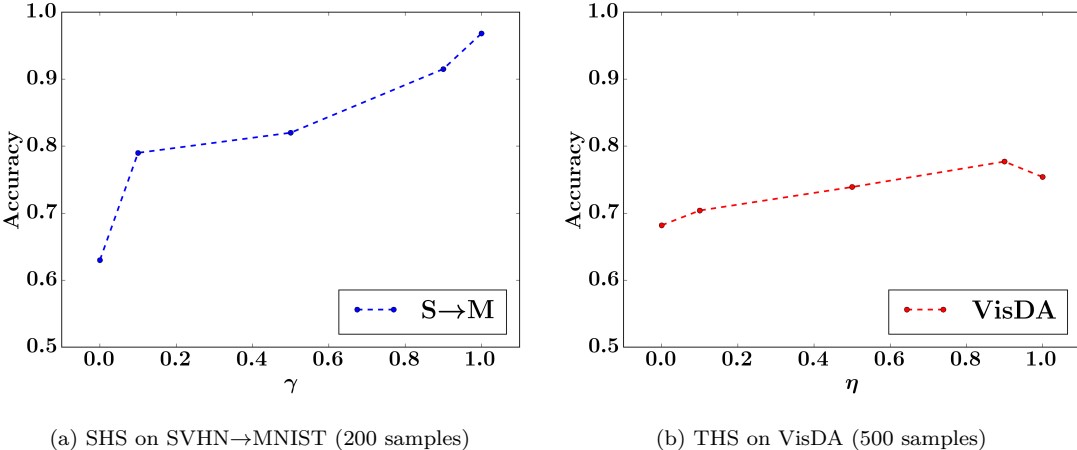

(a) SHS on SVHN→MNIST (200 samples)          (b) THS on VisDA (500 samples)

Figure 14: Validation performance when varying the hyper-parameters $\gamma, \eta$ on a separate dataset

## A.5    Details of the Training Objective

This section briefly explains the training objective of the proposed method. Here we take SHS objective as an example and each term in the objective is clearly defined below:

$$
\begin{cases}
\min_{g,h}(\epsilon_{S_g}(h) + \max_{f_1,f_2} C_{S_g, T_g}(f_1, f_2, h)) \\
\min_{g,f_1,f_2}(\epsilon_{S_g}(f_1) + \gamma \epsilon_{S_g}(f_2))
\end{cases}
\tag{28}
$$

Table 5: Influence of $\lambda$ on the adaptation of SVHN→MNIST under our proposal ($THS + CMD, \eta = 0$).

| SVHN to MNIST | $\lambda = 0.001$ | $\lambda = 0.01$ | $\lambda = 0.1$ |
|---|---|---|---|
| Acc | 98.4±0.1 | 98.6±0.1 | 97.2±0.3 |

where $g$ represent a feature extractor and $h, f_1, f_2$ are classifiers with a softmax output (i.e., $h(x, y)$ represents the confidence of sample $x$ being classified as label $y$). First, we use can the cross-entropy loss as a common surrogate of the source error:

$$\epsilon_{S_g}(h) = -\mathbb{E}_{x,y \sim S} \log(h(g(x), y)) \tag{29}$$

Then we use the proposed cross margin discrepancy to reorganize $C_{S_g, T_g}(f_1, f_2, h)$. Before further derivation, we introduce three proxy distributions $S_{f_1=f_2}, S_{f_1 \setminus f_2}, S_{f_2 \setminus f_1}$ induced by $f_1, f_2$, where $S_{f_1=f_2} = \{x, y | x \sim S, l_{f_1}(x) = l_{f_2}(x) = y\}, S_{f_1 \setminus f_2} = \{x, y | x \sim S, y = l_{f_1}(x) \neq l_{f_2}(x)\}, S_{f_2 \setminus f_1} = \{x, y | x \sim S, l_{f_1}(x) \neq l_{f_2}(x) = y\}$ (the labeling function $l$ defined in main paper Eq.7 returns the most confident prediction). Analogously, we define others like $S_{f_1=h}, S_{f_2=h}, T_{f1 \setminus f2}$, etc. According to the definition of CMD (Sec.3.4):

$$
\begin{aligned}
&\hat{C}_{S_g, T_g}(f_1, f_2, h) \\
&= \mathbb{E}_{x,y \sim S_{f_1=f_2}} [\log \max(f_1(g(x), y), f_2(g(x), y)) + \log \max(1 - f_1(g(x), y), 1 - f_2(g(x), y))] \\
&\quad + \mathbb{E}_{x,y \sim S_{f_1 \setminus f_2}} [\log f_1(g(x), y) + \log(1 - f_2(g(x), y))] + \mathbb{E}_{x,y \sim S_{f_2 \setminus f_1}} [\log(1 - f_1(g(x), y)) + \log f_2(g(x), y)] \\
&\quad + \mathbb{E}_{x,y \sim T_{f_1=f_2}} [\log \max(f_1(g(x), y), f_2(g(x), y)) + \log \max(1 - f_1(g(x), y), 1 - f_2(g(x), y))] \\
&\quad + \mathbb{E}_{x,y \sim T_{f_1 \setminus f_2}} [\log f_1(g(x), y) + \log(1 - f_2(g(x), y))] + \mathbb{E}_{x,y \sim T_{f_2 \setminus f_1}} [\log(1 - f_1(g(x), y)) + \log f_2(g(x), y)] \\
&\quad + \mathbb{E}_{x,y \sim T_{f_1=h}} [\log \max(f_1(g(x), y), h(g(x), y)) + \log \max(1 - f_1(g(x), y), 1 - h(g(x), y))] \\
&\quad + \mathbb{E}_{x,y \sim T_{f_1 \setminus h}} [\log f_1(g(x), y) + \log(1 - h(g(x), y))] + \mathbb{E}_{x,y \sim T_{h \setminus f_1}} [\log(1 - f_1(g(x), y)) + \log h(g(x), y)] \\
&\quad - \mathbb{E}_{x,y \sim S_{f_2=h}} [\log \max(h(g(x), y), f_2(g(x), y)) + \log \max(1 - h(g(x), y), 1 - f_2(g(x), y))] \\
&\quad - \mathbb{E}_{x,y \sim S_{f_2 \setminus h}} [\log f_2(g(x), y) + \log(1 - h(g(x), y))] - \mathbb{E}_{x,y \sim S_{h \setminus f_2}} [\log(1 - f_2(g(x), y)) + \log h(g(x), y)]
\end{aligned} \tag{30}
$$

where part of this objective can be regarded as a CGAN objective aiming to align the conditional distributions of three pairs of hypothesis induced distributions ($S_{f_1=f_2} \Leftrightarrow T_{f_1=f_2}$, $S_{f_1 \setminus f_2} \Leftrightarrow T_{f_2 \setminus f_1}$, $S_{f_2 \setminus f_1} \Leftrightarrow T_{f_1 \setminus f_2}$) under a minor assumption that $f_1, f_2$ are more confident about $S, T$ respectively.

As we explain in A.1.1, in order to avoid unnecessary oscillation during adversarial learning, we only optimize a part of the objective (the part treated as fake by the classifier) w.r.t $g, h$ so that it can be consistent with general GAN objective. This implies the objective ($\hat{C}_{S_g, T_g}(f_1, f_2, h)$) we try to maximize is not equivalent to that ($\check{C}_{S_g, T_g}(f_1, f_2, h)$) we try to minimize:

$$
\begin{aligned}
&\check{C}_{S_g, T_g}(f_1, f_2, h) \\
&= \mathbb{E}_{x,y \sim S_{f_1=f_2}} \log \max(1 - f_1(g(x), y), 1 - f_2(g(x), y)) \\
&\quad + \mathbb{E}_{x,y \sim S_{f_1 \setminus f_2}} \log(1 - f_2(g(x), y)) + \mathbb{E}_{x,y \sim S_{f_2 \setminus f_1}} \log(1 - f_1(g(x), y)) \\
&\quad + \mathbb{E}_{x,y \sim T_{f_1=f_2}} \log \max(1 - f_1(g(x), y), 1 - f_2(g(x), y)) \\
&\quad + \mathbb{E}_{x,y \sim T_{f_1 \setminus f_2}} \log(1 - f_2(g(x), y)) + \mathbb{E}_{x,y \sim T_{f_2 \setminus f_1}} \log(1 - f_1(g(x), y)) \\
&\quad + \mathbb{E}_{x,y \sim T_{f_1=h}} \log \max(1 - f_1(g(x), y), 1 - h(g(x), y)) \\
&\quad + \mathbb{E}_{x,y \sim T_{f_1 \setminus h}} \log(1 - h(g(x), y)) + \mathbb{E}_{x,y \sim T_{h \setminus f_1}} \log(1 - f_1(g(x), y)) \\
&\quad - \mathbb{E}_{x,y \sim S_{f_2=h}} \log \max(1 - h(g(x), y), 1 - f_2(g(x), y)) \\
&\quad - \mathbb{E}_{x,y \sim S_{f_2 \setminus h}} \log(1 - h(g(x), y)) - \mathbb{E}_{x,y \sim S_{h \setminus f_2}} \log(1 - f_2(g(x), y))
\end{aligned} \tag{31}
$$

Since we apply different measurements to the source error and the discrepancy, we introduce a scaling factor $\lambda$ to ensure that neither of them can dominate the back-propagation. $\lambda = 0.01$ is used in all experiments because we observe that source error is usually around $1e - 2$ and the discrepancy is usually around $1e0$. We also conduct a simple experiment to check the influence of $\lambda$ and report the results in Tab.6.

Table 6: The accuracy of ResNet-50 model fine-tuned on the Office-Home dataset. * represents the imbalance label distribution setting where we manually remove the samples of an entire class from the target domain.

| METHOD | Ar→Cl | Pr→Cl | Rw→Cl |
|---|---|---|---|
| Source Only | 34.9 | 31.2 | 41.2 |
| DANN(Ganin et al., 2016) | 45.6 | 43.7 | 51.8 |
| MCD(Saito et al., 2017b) | 51.9 | 51.6 | 55.8 |
| CDAN(Long et al., 2018) | 50.7 | 50.9 | 56.7 |
| SymNets(Zhang et al., 2019a) | 47.7 | 48.8 | 52.6 |
| SPL(Wang & Breckon, 2020) | 54.5 | 53.1 | 55.3 |
| AADA(Yang et al., 2020) | 54.0 | 51.8 | 57.4 |
| SRDC(Tang et al., 2020) | 52.3 | 53.8 | 57.1 |
| SCAL(Wang et al., 2022) | 55.3 | 51.6 | 57.8 |
| ours $(THS + CMD, \eta = 0.9)$ | 60.3 | 59.2 | 62.7 |
| DANN* | 43.1 | 40.5 | 49.2 |
| MCD* | 50.5 | 49.7 | 54.3 |
| ours* $(THS + CMD, \eta = 0.9)$ | 60.0 | 58.5 | 62.2 |

## A.6 Imbalance Label Distribution

In this section, we conduct an additional experiment on the situation where the label distribution of source is very different from that of target domain. We choose Office-Home dataset and manually remove the samples of an entire class ('clock') from the target (Clipart) domain to create an imbalance scenario. We report the adaptation accuracy in Tab.6. In theory, our proposal does not suffer from the imbalance label distribution. Because unlike other methods that can only match the marginal distributions, our proposal is designed to align the conditional distributions. In practice, the experimental results demonstrate our theory since there is no remarkable performance drop in our proposal compared to other methods. Besides, despite this extreme imbalance label distribution case, our proposal still outperforms other methods that trained on the full target domain.

## A.7 Ablation Study

In this section, we conduct a simple ablation study to show how each part of the proposal, i.e., the objective including the joint error, the cross margin discrepancy and the pseudo labels induced hypothesis space, contributes to the performance gain respectively. For comparisons, we choose MCD(Saito et al., 2017b) as a baseline since its assumption about the hypothesis space is similar to our SHS proposal. Besides, the two methods share the same network architecture in implementation, which makes the experimental results directly comparable. From Tab. 7, it can be seen that the 4th row of our method shows the effectiveness of the proposed upper bound, where we use $L_1$ as the discrepancy measurement without involving any pseudo target information and improve the performance compared to MCD in the 1st row. The 6th row shows the result when replacing it with the cross margin discrepancy. The last row shows the result when leveraging the information from pseudo-labeled target samples to construct a more reliable hypothesis space for $f_2$. These results show that every part of our proposal helps to improve the performance. In order to further verify the effectiveness of our proposal, we conduct another ablation study on VisDA dataset. The results in Tab. 8 reveal a similar conclusion that each part of our proposal is beneficial to the adaptation despite a more complicated scenario.

Table 7: A simple ablation study to reveal the contribution from each part of the proposal ($\sqrt{}$ in the column "optimal joint error" means to include the joint error in the upper bound, in contrast to the training objective proposed by MCD; $\sqrt{}$ in the column "cross margin discrepancy" means to use the proposed discrepancy measurement for $\epsilon$, whereas $\times$ indicates we use $L_1$ norm instead; $\sqrt{}$ in the column "pseudo labels induced hypothesis space" means the hypothesis space for $f_2$ is built based on pseudo labels of target samples, which is corresponding to our THS proposal in the main paper). We repeat the adaptation from SVHN to MNIST 5 times and report the average and the standard deviation of the accuracy.

| METHOD | OPTIMAL JOINT ERROR | CROSS MARGIN DISCREPANCY | PSEUDO LABELS INDUCED HYPOTHESIS SPACE | SVHN TO MNIST |
|---|---|---|---|---|
| BASELINE | $\times$ | $\times$ | $\times$ | 96.2±0.4 |
| | $\times$ | $\sqrt{}$ | $\times$ | 96.6±0.2 |
| | $\times$ | $\times$ | $\sqrt{}$ | 97.1±0.3 |
| | $\times$ | $\sqrt{}$ | $\sqrt{}$ | 97.6±0.2 |
| OURS | $\sqrt{}$ | $\times$ | $\times$ | 96.8±0.2 |
| | $\sqrt{}$ | $\sqrt{}$ | $\times$ | 97.5±0.2 |
| | $\sqrt{}$ | $\times$ | $\sqrt{}$ | 98.2±0.2 |
| | $\sqrt{}$ | $\sqrt{}$ | $\sqrt{}$ | 98.6±0.1 |

Table 8: A simple ablation study to reveal the contribution from each part of the proposal ($\sqrt{}$ in the column "optimal joint error" means to include the joint error in the upper bound, in contrast to the training objective proposed by MCD; $\sqrt{}$ in the column "cross margin discrepancy" means to use the proposed discrepancy measurement for $\epsilon$, whereas $\times$ indicates we use $L_1$ norm instead; $\sqrt{}$ in the column "pseudo labels induced hypothesis space" means the hypothesis space for $f_2$ is built based on pseudo labels of target samples, which is corresponding to our THS proposal in the main paper). We repeat the adaptation on VisDA dataset 3 times and report the average and the standard deviation of the accuracy.

| METHOD | OPTIMAL JOINT ERROR | CROSS MARGIN DISCREPANCY | PSEUDO LABELS INDUCED HYPOTHESIS SPACE | VisDA |
|---|---|---|---|---|
| BASELINE | $\times$ | $\times$ | $\times$ | 71.9±0.4 |
| | $\times$ | $\sqrt{}$ | $\times$ | 73.1±0.3 |
| | $\times$ | $\times$ | $\sqrt{}$ | 76.2±0.3 |
| | $\times$ | $\sqrt{}$ | $\sqrt{}$ | 76.8±0.2 |
| OURS | $\sqrt{}$ | $\times$ | $\times$ | 73.4±0.3 |
| | $\sqrt{}$ | $\sqrt{}$ | $\times$ | 76.4±0.3 |
| | $\sqrt{}$ | $\times$ | $\sqrt{}$ | 79.2±0.2 |
| | $\sqrt{}$ | $\sqrt{}$ | $\sqrt{}$ | 81.6±0.2 |

