# OpenReview forum: "Unsupervised Domain Adaptation via Minimized Joint Error"
_TMLR — Rejected by TMLR_

### Review · Reviewer_pmJu · 2022-09-24

**Summary Of Contributions:**

This paper addressed an important issue in representation based unsupervised domain adaptation. The conventional (representation learning based) domain adaptation approaches would induce a high joint optimal risk. In contrast, this paper aims to upper bound the joint risk and proposed a novel algorithm to address this issue. In the experimental part, this paper demonstrated a strong practical improvement (~30%) in MNIST->SVHN, which is believed to be a challenging task in conventional domain adaptation.


**Requested Changes:**

I would suggest a revision on the math illustrations. This paper has several interesting insights, while the theoretical derivations seem quite unclear and bit confusing during the review. Rigorous and clear presentation could significantly improve the manuscript.

**Strengths And Weaknesses:**

Overall, this paper aims to address an important problem in domain adaptation: the joint optimal risk is not necessarily small when learning a marginal invariant representation. This paper further analyzed the joint optimal risk by upper bounding as a novel prediction loss. The empirical results seem promising. Simultaneously, I think there are several important math issues to be addressed. My comments, which are based on TMLR guidelines, are as follows:

### Would some individuals in TMLR's audience be interested in the findings of this paper?

**YES**.

- Representation learning in the context of unsupervised DA is important. Particularly the theoretical analysis in the joint optimal term.
-  A novel practical algorithm inspired by the theorem.
- Empirical validations are conducted to justify the claims, particularly in the task MNIST-> SVHN.

### Are the claims made in the submission supported by accurate, convincing and clear evidence?


**Partially. Math derivations seem unclear or problematic. Improvements/revisions are required.***
- Equation (1-2). The upper bound changes the optimal predictors. In fact, the optimal predictor in target risk should be $f_T$ (this is obvious). While by deriving the upper bound, the optimal predictor has been changed to $f_S$, this is problematic. Since the optimal predictor is the target domain should be the target labelling function rather than source labelling function. I would think it is the problem of triangle inequality, whereas the equal condition changes.
- Equation (5), the notation $g(S), g(T)$ are quite wired. Upon reflection, I understand it is the representation function on unlabelled source and target domain, but this requires revision.
- Equation (8) and the above paragraph, what does it mean by
$H_2\leq H_f$? Please note $H$ is the hypothesis family, I could not understand that one hypothesis set is smaller than another.
- Equation (9-10), I could understand the constraint on $f_1,f_2$ by ensuring they are informative. But why training these two predictors separately, it would be fine-tuned by $h$ to fast-adapt $f_1$ and $f_2$, right? Besides, the role of $\gamma$ seems a bit confusing for me…
- Equation (11-15), do these losses satisfy the triangle inequality for equation (1)? I feel like the margin loss is not satisfying these properties.
- Eq(18), it is not a min-max problem in my eyes. Actually the loss in (18) is defined as $\log  \max(f_1(x),f_2(x))$, this is different from $\max \log f(x)$.
- [Minor] I would suggest authors to integrate Appendix A4 into the main manuscript, this could enable a better understanding in related works.

---

> ### Author Response · Authors · 2022-09-28
> **Answers to Reviewer pmJu**
>
> Thanks for your valuable comments.
>
> We response to your concerns as follows.
>
> 1.Indeed the optimal solution for $h$ is changed from $f_T$ to $f_S$ due to the triangle inequality, but it is not a serious problem in our opinion.
>
> First of all, we can prove that all the upper bound based methods change the optimal solution as:
> $\epsilon_T(h) \leq \epsilon_T(h,h') + \epsilon_T(h',f_T)$
>
> Here, by introducing the triangle inequality, the optimal solution for $h$ becomes $h'$ and it is inevitable.
> For example, when we take a look at the upper bound proposed by Ben-David:
> $\epsilon_T(h) \leq \epsilon_S(h) + \sup_{f_1,f_2 \in H}| \epsilon_S(f_1,f_2)- \epsilon_T(f_1,f_2))| + \lambda$,
> we can see the optimal solution for $h$ is also $f_S$
>
> Secondly, we think it is difficult to directly approximate $f_T$ using $h$ in practice with out any accurate labeled target samples. And it is why most of he upper bound based methods try to seek other solutions for $h$. In our algorithm, the proposed upper bound can jointly align the distributions of the two domains and minimize the joint error, which makes $f_S$ a feasible approximation for $f_T$.
>
>
> 2.We will revise the representation of extracted feature space with: $S_g=${ $(g(x_s),y_s)|(x_s,y_s)\sim S$}, $T_g=${$g(x_t)|x_t\sim T$}
>
> 3.We use $H_2 \leq H_f$ to represent that $H_2$ is a subspace of $H_f$. We find this notation in some other papers and we are not sure if there is any more common notation for this relation.
>
> 4.The idea of training two classifiers is inspired by MCD which originates from Ben-David's theory.
>
> Except for the common role to relax the upper bound with the supremum term, $f_1,f_2$ have other important roles in our algorithm unlike $h$ which only aims to minimize the objective.
>
> For simplicity, we assume the predictions of $f_1, f_2$ agree.
>
> $f_2$ can leverage the pseudo predictions given by $h$ which makes it more discriminative than $f_1$ on the target domain.
>
> $f_1$ is usually more discriminative than $f_2$ on the source domain because of the weight $\gamma, \eta$.
>
> Given the definition of CMD, a part of our objective can be written as:
>
> $\epsilon_S(f_1,f_2)+\epsilon_T(f_1,f_2)=E_{x\sim S}[\log (f_1(x))+\log (1-f_2(x))] + E_{x\sim T}[\log (f_2(x))+\log (1-f_1(x))]$
>
> Our proposal resembles GAN objective where $f_1,f_2$ act as domain classifiers that treat the source and target domain as true data respectively.
>
> As for the role of $\gamma$:
>
> In theory, we hope to build a hypothesis space with classifiers that can predict the source domain with an accuracy of $\gamma$ as we think not all the source samples help to recognize the target domain especially when the domain gap is huge.
>
> In practice, as explained above, a weighted source risk for $f_2$ will lead to a less discriminative classifier for source domain compared to $f_1$, which gives a more appropriate GAN-like objective.

---

> ### Author Response · Authors · 2022-09-28
> **Answers to Reviewer pmJu**
>
> 5.Before we answer this question, we want to introduce a more general problem associated with the consistency between the algorithm and theory.
>
> We take the Ben-David's theory as an example:
>
> \begin{align}
> \epsilon_T(h)
> & \leq \epsilon_T(h^\ast) + \epsilon_T(h,h^\ast)\nonumber\\\\
> & \leq \epsilon_T(h^\ast) + \epsilon_S(h,h^\ast) +  |\epsilon_T(h,h^\ast) - \epsilon_S(h,h^\ast)|\nonumber\\\\
> & \leq \epsilon_T(h^\ast) + \epsilon_S(h^\ast) +  \epsilon_S(h) + |\epsilon_T(h,h^\ast) - \epsilon_S(h,h^\ast)|\nonumber\\\\
> & = \epsilon_S(h) + |\epsilon_T(h,h^\ast) - \epsilon_S(h,h^\ast)| + \lambda\nonumber\\\\
> & \leq \epsilon_S(h) + \sup_{h,h' \in H}| \epsilon_S(h,h')- \epsilon_T(h,h'))| + \lambda\nonumber
> \end{align}
>
> From the derivation, we know the triangle inequality is essential to build the theory. Besides, the measurement of source error and the measurement of the terms related to marginal discrepancy should be the same. However, most of the upper bound based methods violate these rules which is known as the gap between the algorithm and theory. For instance, MCD chooses cross entropy for the source error but replaces the marginal discrepancy with a $L_1$ norm between the predictions of two classifiers. As for MDD, it uses logistic loss as a surrogate to approximate 0-1 loss which no longer satisfies the triangle inequality.
>
> In conclusion, any upper bound based method that uses cross entropy to measure the source error must violate the consistency requirement.
>
>
> Now we come back to the question. Indeed our loss function does not satisfy the triangle inequality, however we can prove that given the condition that $f_1,f_2,f_3$ agree on $x$, the primitive form of our loss (Eq.17) meets the consistency requirement:
>
> $d(f_1,f_2,x)=|\log f_1(x,y)- \log f_2(x,y)|$
>
> It is trivial that the above loss function satisfies the triangle inequality since $d(f_1,f_2,x)\leq d(f_1,f_3,x)+d(f_3,f_2,x)$
>
> As for the source error, we can rewrite it as:
>
> $\epsilon_S(h)=E_{x,y\sim S}|\log f_S(x,y)- \log h(x,y)|=-E_{x,y\sim S}\log h(x,y)$
> ,where the cross entropy is proved to be a special case of our proposal.
>
> As the training proceed, the distributions of source and target domain will be eventually
> aligned. This means $f_1,f_2,h$ will finally agree on most of the samples and we can conclude that our proposal asymptotically satisfies the consistency requirement.
>
>
> 6.The optimization is a min-max game because in Eq.7:
>
> $\min_{g\in H_g,h\in H_f}(\epsilon_{g(S)}(h) + \max_{f_1,f_2\in H_f}C_{g(S),g(T)}(f_1,f_2,h))$
>
> we maximize $C_{g(S),g(T)}(f_1,f_2,h)$ w.r.t $f_1,f_2$ and minimize the same objective w.r.t $g,h$
>
> Eq.18 is the definition of the proposed loss function CMD, which has noting to do with the min-max optimization.
>
> 7.We will see if we could put A.4 into the main body under the page limit.

---

> > ### Comment · Reviewer_pmJu · 2022-11-01
> > **Discussions**
> >
> > Thanks for your response. Several concerns have been properly addressed, while I feel difficulty in understanding the following.
> >
> > > Secondly, we think it is difficult to directly approximate f_t  using h in practice without any accurate labeled target samples. And it is why most of the upper bound based methods try to seek other solutions for h. In our algorithm, the proposed upper bound can jointly align the distributions of the two domains and minimize the joint error, which makes f_s a feasible approximation for f_t.
> >
> > I could not understand *f_s being a feasible approximation for f_t.* What is the assumption within your paper? why f_s could approximate f_t?

---

> > > ### Author Response · Authors · 2022-11-01
> > > **Answers to Reviewer pmJu**
> > >
> > > Thanks for your comment.
> > >
> > > We response to your concern as follows.
> > >
> > > A common assumption for distribution alignment based domain adaptation is that if we align the marginal distributions of source and target domain, then a classifier for the source domain can approximately classify the target domain. This idea is proposed by Ben-David (and widely used in DANN, ADDA, etc.), which is supported by the following upper bound:
> > > $\epsilon_T(h,f_T) \leq \epsilon_S(h,f_S) + \sup_{f_1,f_2 \in H}| \epsilon_S(f_1,f_2)- \epsilon_T(f_1,f_2))| + \lambda$
> > >
> > > Here, the optimal solution for $h$ is $f_S$.  The reason why $f_S$ becomes a feasible approximation for $f_T$ is that the supremum term can align the marginal distributions which brings $f_S,f_T$ close to each other.
> > >
> > > In our algorithm,  the optimal solution for $h$ is also $f_S$. However,  the difference is that we align the conditional distributions of source and target domain using pseudo labels (A.5, Paragraph 2).  The problem of marginal distribution alignment is that a classifier trained on the source domain does not necessarily classify the target domain due to the potential overlap of different classes (Fig.1b). In contrast, if the conditional distributions are aligned, a classifier for the source domain must classify the target domain.

---

> > > > ### Comment · Reviewer_pmJu · 2022-11-02
> > > > **Further questions**
> > > >
> > > > Thanks for your quick response. While I would think the inherent assumption should be the small joint optimal risk $\lambda$ on two domains. Surely f_T \in H is one possible assumption and conditional alignment could potentially address this.
> > > >
> > > > However, there is the problem of identification, i.e since the target domain Y is unknown, thus a perfect conditional matching is quite difficult. What kind of assumptions (in general, not theoretical) are you made within your method here ?
> > > >
> > > > I am sorry for repeating asking the assumption since this paper proposed novel theories.  I feel that it is a bit hard to understand  assumptions behind the theory.

---

> > > > > ### Author Response · Authors · 2022-11-03
> > > > > **Answers to Reviewer pmJu**
> > > > >
> > > > >
> > > > > Thanks for your comment and we are glad to discuss this further.
> > > > >
> > > > > The algorithms proposed by Ben-David and its extensions (e.g., DANN, ADDA) assume a small joint error $\lambda$ because they cannot directly optimize the joint error.
> > > > >
> > > > > In contrast, we build an upper bound over the joint error (Eq.2, Eq.3) and minimize it by aligning the conditional distributions (A.5). Therefore, we do not think it is correct to call this an assumption.
> > > > >
> > > > > Besides, we do not need to assume $f_T \in H$ because the theory still holds by replacing $f_T$ with its approximation inside the hypothesis space $f^{\ast}_T \in H$ (Sec.3.2, Paragraph 1 and A.3).
> > > > >
> > > > > To make the proposed upper bound (Eq.1) optimizable, we need the following inequality to hold:
> > > > > $C_{S_g,T_g}(f^\ast_S,f^\ast_T,h) \leq \max_{f_1 \in H_1, f_2 \in H_2}C_{S_g,T_g}(f_1,f_2,h)$ (In Eq.8, we omit $\ast$ for simplicity)
> > > > >
> > > > > A sufficient condition for this is $f^\ast_S \in H_1, f^\ast_T \in H_2$.
> > > > >
> > > > > Building a hypothesis class $H_1$ that contains $f^\ast_S$  can be easily achieved since we have source labels.
> > > > > However, as you mentioned, it is difficult to build a hypothesis class $H_2$ that contains $f^\ast_T$ without increasing its size. In spite of the effort we make to relive the influence caused by wrong pseudo labels (e.g., using a large $\eta$ in Eq.10), we cannot guarantee the above inequality especially when the domain gap is huge.
> > > > >
> > > > > Therefore, if there is an assumption (necessary condition) for the proposal, it should be:
> > > > >
> > > > > with a hypothesis class $H_2$ built on pseudo labels that may not contain $f^\ast_T$, the following inequality still holds:
> > > > > $C_{S_g,T_g}(f^\ast_S,f^\ast_T,h) \leq \max_{f_1 \in H_1, f_2 \in H_2}C_{S_g,T_g}(f_1,f_2,h)$
> > > > >
> > > > > This assumption is proved to be acceptable judging from the large performance gain in those hard adaptation scenarios where the domain gap is huge (e.g., AR to CL, PR to CL, RW to CL in OfficeHome, Tab.3). The result implies that in the hard scenarios, even if the hypothesis class $H_2$ is built on many wrong pseudo labels, the objective we keep minimizing is still an upper bound for the joint error.

---

> > > > > > ### Comment · Reviewer_pmJu · 2022-11-03
> > > > > > **Thanks**
> > > > > >
> > > > > > Thanks for further explanations.  The inherent assumption seems making sense to me.
> > > > > >
> > > > > > -----------------
> > > > > > Minor point: About *the objective we keep minimizing is still an upper bound for the joint error.* This is true, while the achievability and tightness of the upper bound are also crucial. We could easily create any sort of arbitrary upper bound by (for example)
> > > > > >
> > > > > > $$ L(x) \leq 100000 * L(x) +  g(x) $$
> > > > > >
> > > > > > where $g(x)$ is an arbitrary function. It is still an upper bound and I could further create infinity upper bounds. While only the achievable and tight bounds make more sense.

---

> > > > > > > ### Author Response · Authors · 2022-11-03
> > > > > > > **Answers to Reviewer pmJu**
> > > > > > >
> > > > > > > As you mentioned, the achievability and tightness of the upper bound are both crucial.
> > > > > > >
> > > > > > > However, we think there is a trade-off between the achievability and tightness.
> > > > > > >
> > > > > > > In our algorithm, the tightness of the upper bound is given by the Rademacher complexity (A.2), which is closely related to the size of the hypothesis space. This is the reason why we prefer a constrained hypothesis space $H_2$ that may be built on some wrong pseudo labels rather than an unconstrained space that can guarantee the achievability, even if it may increase the risk of violating the achievability.

---

### Review · Reviewer_NxJp · 2022-09-26

**Summary Of Contributions:**

The method gives a hypothesis space induced by source/pseudo-target labels to tighten up the proposed objective to reduce searching space. A discrepancy measurement, cross margin discrepancy (CMD), measures the dissimilarity between hypotheses. The method borrows from the upper bound guarantee of Ben-David et al., and extrapolates on the benefit of joint distribution in alignment enhancement of distribution matching by also implicitly constraining the gradient function to be a smooth bell curve rather than a nonlinear/stepwise function. Such constraints could retain alignment directions given a coarse initialization on the distribution matching (like k-means, or hard margin), and retains the decision boundary by changing the approximation given by the gradients of the feature embedding function. The loss is formulated from maximum mean discrepancy and based on the analysis of the discrepancy as the mixture of the sum of the margin loss of the source-driven hypothesis space and target-driven hypothesis. Some experiments demonstrate the performance of the proposed approach.

**Broader Impact Concerns:**

No concern in the categories that would involve ethical implications.

**Requested Changes:**

Besides the weaknesses, there are some minor comments as below:

Eq. 6,7 - not clear what are $f_1$ and $f_2$, if the results are summarized from the appendix, should be put in the main body.
What is $H_f$
In eq. 9, eq. 10. how are $h$ and $f_1, f_2$ related? Eq. 4 and Eq. 5 do not tell me what $f_1, f_2$ are.
Eq. 14 - disagreement in distribution, please provide the reference.
Eq. 15 - where is the softmax in the equation?
Eq. 16 is not clear. What are the tricks in eq 16? How does it mitigate the burden of vanishing? Please provide a reference.
What is the primitive form of in eq. 17, of what? $l_{f1}(x) = l_{f2}(x) = y$? please break down this or reference it in the appendix.
Why is eq. 18 the dual form? Not clear at all.

Also did not include in the comparison, another popular approach GAN/VAE-based baselines for VAEs, that would give a nominal understanding of the advantages of the methods. The type of comparison that would help to contrast is the performance on ambiguous numbers, ie. 4,7.

It is not clear what the failure case could be for the assumption, for the imbalanced cases, some inclusion of the data imbalance (even if artificially induced), would be helpful.


**Strengths And Weaknesses:**

Strengths:
1. The proposed approach models the upper bound of joint error in domain transfer.
2. The proposed approach contains the ways of estimating functional spaces.
3. Empirical results seem good.

Weaknesses:
1. The proposed approach is based on some assumptions, but there is no discussion on whether such assumptions hold in practice.
2. it is not clear whether Eq. 2 holds. Is $\epsilon$ a distance measure?
3. I do not understand Eq. 9 and Eq. 10.
4. The implementation is not clear. The authors do not release the code.
5. It is not clear whether the hyperparameters are stable across different datasets.

---

> ### Author Response · Authors · 2022-09-28
> **Answers to Reviewer NxJp**
>
> Thanks for your valuable comments.
>
> We response to your concerns as follows.
>
> 1.Could you please make your question more specific because we are not sure what assumption your are referring to.
>
> We never make any assumptions like balanced dataset.
>
> 2.$\epsilon$ is a distance metric. (Page 3, Section 3.1, Paragraph 1, Line 5)
>
> 3.Eq.9 and Eq.10 are two training objectives which have different hypothesis space constraints.
>
> The difference is whether to leverage the pseudo predictions given by $h$.
>
> We are willing to provide more details if you could make your question more specific.
>
>
> 4.The details of the experimental settings are provided in A.1 and the training procedure is Sec.3.5 and A.5.
>
> Currently, we have no plan to release the code of an unpublished work.
>
> However, if the code is essential to secure your recommendation for acceptance, we can send it to you in private.
>
> 5.The hyper-parameter $\eta$ is stable across different datasets.
>
> When the domain gap is huge, e.g. VisDA (Table 2), OfficeHome (Table 3), Office31 (Table 4), $\eta=0.9$ gives reliable performance.
>
> When the domain gap is small, e.g. Digit dataset, the accuracy remains high and varies subtly for different value of $\eta$. (Section 4.1, Paragraph 2, Line 6)

---

> ### Author Response · Authors · 2022-09-28
> **Answers to Reviewer NxJp**
>
> As for the minor comments:
>
> 1.In Eq.6:
> \begin{align}
> \epsilon_T(h)
> & \leq \epsilon_S(h) + C_{S,T}(f_S,f_T,h)\nonumber\\\\
> & \leq \epsilon_S(h) + \sup_{f_1,f_2\in H}C_{S,T}(f_1,f_2,h)\nonumber
> \end{align}
>
> $f_1,f_2$ are just two hypotheses that could maximize $C_{S,T}$ such that we can build an upper bound and optimize the intractable term $C_{S,T}(f_S,f_T,h)$.
>
> 2.$H_f$ is a subspace of $H$
>
> 3.$f_2$ can leverage the pseudo predictions given by  $h$, which makes it more discriminative than  $f_1$ on the target domain.
>
>  $f_1$ is usually more discriminative than  $f_2$ on the source domain because of the weight $\gamma, \eta$ .
>
> 4.Eq.4 and Eq.5 describe the theory proposed by Ben-David.
>
> $f_1,f_2$ are just two hypotheses that could maximize the marginal discrepancy term, noting more.
>
> 5.Eq.16 and Eq.17 show that our proposed loss CMD can be interpreted as the sum of two margin loss.
>
> As for the definition of margin loss, we refer to Koltchinskii & Panchenko (2002) in Section 3.4, Paragraph 2, Line 3.
>
> 6.$f$ is a multi-layer perceptron whose output layer is a $softmax$ function. (Section 3.4, Paragraph 3, Line 2)
>
> 7.The trick in Eq.18 is to maximize $\log (1-f(x))$ instead of $-\log f(x)$, which is introduced in  Goodfellow et al. (2014).
>
> When $f(x)$ is close to 0, the gradient of $-\log f(x)$ will explode.
>
> 8.Eq.19 is the primitive definition of CMD when $l_{f_1}(x)=l_{f_2}(x)=y$.
>
> We call Eq.20 a dual form because we optimize $\log (1-f(x))$ instead of $-\log f(x)$.
>
> 9.We provide the performance of several GAN-based methods like ADDA, DANN, CDAN for comparisons.
>
> As for VAE-based methods, if you are referring to the image-to-image translation methods like UNIT, we have to say those methods usually have much worse performance compared to
> the methods based on error bound like ours. Because those methods focus on the quality of transformed images while ours optimize the upper bound of the target error which is directly related to the classification accuracy. E.g., the adaptation accuracy of SVHN to MNIST given by UNIT is 0.91. Besides, those methods usually cannot apply to complicated dataset like VisDA due to the huge translation cost.
>
> 10.We never make any assumptions like balanced dataset, and our proposal does not suffer from the imbalance label distribution in theory. Because unlike other methods that can only match the marginal distributions, our proposal is designed to align the conditional distributions (A.5, Eq.30). In fact , VisDA is an imbalance dataset where the 'car' class has 10000 samples　while the 'knife' class only has 2000 samples, and our proposal works just fine. We can conduct an additional experiment on imbalance dataset by manually removing an entire class from the target domain if it helps to secure your recommendation for acceptance.

---

### Review · Reviewer_chqD · 2022-10-17

**Summary Of Contributions:**

This paper tackles the problem of unsupervised domain adaptation. They are motivated by the issue that the joint error is usually ignored due to the intractability by previous works. To tackle this problem, they propose a newobjective that relates to an upper bound of the joint error, adopt source/pseudo-target labels, and introduce a cross-margin discrepancy to alleviate the instability during adversarial learning. Proposed method shows improvement on several UDA benchmarks.

**Requested Changes:**

I suggested the author provide：
1. a concrete algorithm flow
2. Move figure 14 to the main text with modification. E.g, what the two constructed distributions are? Modify the equation with a more simple expression. Please refer the figure 3 from the MCD paper. Figure 14 (b) should be maximized not minimized.
3. The connection between the proposed upper bound and Cross Margin Discrepancy are not clearly elaborated. E.g, in figure 14, the eqns are from sec 3.1 while the optimization is actually based on sec 3.3.


**Strengths And Weaknesses:**

Pros:
+The paper is well motivated by the joint error and author derive the upper-bound for it.
+Show good empirical results relative to existing work, especially on Office-home

Cons:
-The method overclaims the contribution on targeting the joint error. e.g, [1,2] this issue has been explored several years ago. The author claim that methods based on Ben-David's bound cannot solve the joint error issue while many subsequent works made efforts to reduce the joint error term within this formula.
-The optimization procedure on “Cross Margin Discrepancy” is very hard to follow, even looking at the supplementary part. And the whole design is very much like the three-classifier version of MCD.
-The reason why this paper makes the readers hard to follow is that: the optimization/implementation and their proposed theory are not elaborated well, yet the figures in the main text do not help the explanation and pseudo code/code is not provided. It raised up the understanding and reproducibility of the work.

[1] Learning Semantic Representations for Unsupervised Domain Adaptation, ICML18
[2] Transferability vs. Discriminability: Batch Spectral Penalization for Adversarial Domain Adaptation, ICML19

---

> ### Author Response · Authors · 2022-10-18
> **Answers to Reviewer chqD**
>
> Thanks for your valuable comments.
>
> We response to your concerns as follows.
>
> 1.We add Sec.3.5 in the main body to describe the training procedure.
>
> 2.We move Fig.14 to Sec.3.5 and revise some expressions.
>
> In A.5, we introduce some proxy distributions $S_{f_1=f_2},S_{f_1\backslash f_2}, S_{f_2\backslash f_1}$ induced by $f_1,f_2$ such that we can rewrite the training objective and interpret it with CGAN.
>
> - $S_{f_1=f_2}=${$x,y|x\sim S, l_{f_1}(x)=l_{f_2}(x)=y$} represents a distribution where $y$ is labeled jointly by $f_1,f_2$ when they agree
> - $S_{f_1\backslash f_2}=${$x,y|x\sim S, y=l_{f_1}(x)\neq l_{f_2}(x)$} represents a distribution where $y$ is labeled by $f_1$ when $f_1,f_2$ disagree
> - $S_{f_2\backslash f_1}=${$x,y|x\sim S, l_{f_1}(x)\neq l_{f_2}(x)=y $} represents a distribution where $y$ is labeled by $f_2$ when $f_1,f_2$ disagree
>
> $\min$ is correct for Fig.14 (Fig.4) because $-\min_{f_1,f_2} C_{S_g,T_g}(f_1,f_2,h)$ is equivalent to $\max_{f_1,f_2} C_{S_g,T_g}(f_1,f_2,h)$.
>
> 3.We propose the joint error based upper bound in Sec.3.1, Eq.1 (here $\epsilon$ can be any distance metric):
> - $\epsilon_T(h) \leq \epsilon_S(h) + \epsilon_T(f_S,f_T) + \epsilon_S(f_S,f_T) + \epsilon_T(h, f_S) - \epsilon_S(h, f_T)$
>
> Then we define the actual loss function of $\epsilon$ with cross margin discrepancy in Sec.3.4, Eq.15:
> - $\epsilon_D(f_1,f_2)=E_{x\in D}[cmd(f_1,f_2,x)]$

---

> ### Author Response · Authors · 2022-10-19
> **Comments on the Drawbacks mentioned by Reviewer chqD**
>
> The reviewer claims that we exaggerate the contribution on the joint error, but we do not agree.
>
> Generally, the marginal distributions of source and target domain can be perfectly aligned with the help of GAN-based framework. However, the adaptation accuracy given by the existing methods is far from perfect especially when the domain gap is large and it is caused by the wrong alignment between the conditional distributions. This means after the alignment, samples from different classes in source and target domain are mixed in the same cluster, which leads to an inevitable large joint error caused by the undesired overlap in the feature space. We explain it in Eq.5 and this problem can never be totally addressed even with our proposal.
>
> The contribution of this work is that we construct a theoretical upper bound on the joint error that can be actually minimized, which we believe is novel in domain adaptation. The effectiveness is supported by substantial evidence (e.g., AR to CL, PR to CL, RW to CL in OfficeHome) which shows that our proposal has a great advantage especially when the domain gap is large.
>
> As for the previous work [1] mentioned by the reviewer:
> - Theoretically, in addition to the source error and the marginal discrepancy, [1] adds a loss into the objective to align labeled source centroid and pseudo-labeled target centroid. They claim the proposal can minimize the joint error, but we are not convinced. In Eq.7 of [1], the joint error $C$ is bounded by:
> $C \leq \min_{h \in H} \epsilon_S(h,f_S)+\epsilon_T(h,f_S)+\epsilon_T(f_S,f_{\hat{T}})+\epsilon_T(f_T,f_{\hat{T}})$. We understand the fist three terms can be minimized based on their algorithm, but the last term $\epsilon_T(f_T,f_{\hat{T}})$ is not well-explained. Actually, the last term represents the error rate of the predictor $f_{\hat{T}}$, which is bounded by the sum of the source error, marginal discrepancy and the joint error. It seems that they use the joint error to bound the joint error which we think is problematic.
>
> - Experimentally, our method outperforms [1] by __6.9%__ on the accuracy of SVHN to MNIST.
>
> - Besides, [3] also tries to align labeled source centroid and pseudo-labeled target centroid, but they never mention their proposal can reduce the joint error.
>
> [3] Unsupervised Domain Adaptation via Structurally Regularized Deep Clustering, CVPR 2020
>
>
> As for the previous work [2] mentioned by the reviewer:
>
> - Theoretically, [2] applies a stricter match between the marginal distributions by penalizing the eigenvectors with largest singular values in the feature representations, such that the other eigenvectors with relatively smaller singular values can be matched instead of being suppressed. However, it is not well explained in theory why this can reduce the joint error, which is a problem more related to the conditional discrepancy. We agree the algorithm can lead to a discriminative (well-clustered) feature space where the ratio of between-class variance to within-class variance is maximized, but it cannot guarantee the target samples are placed in the correct clusters. Once the target samples are placed in the wrong cluster, a larger joint error is inevitable.
>
> - Experimentally, our method outperforms [2] by __5.7%__ on the average accuracy of OfficeHome dataset.
> Besides, in the scenarios where the domain gap is huge (e.g., AR to CL, PR to CL, RW to CL in OfficeHome), [2] shows no advantage over other methods,  which cannot support their claim on controlling the joint error.
>
> ***
> The difference between our proposal and MCD is described in Sec.3.3.2.
> In short, under some additional assumptions, our proposal can reduce to MCD which means our upper bound is designed in a more general way:
>
> - MCD is still based on Ben-David's upper bound which ignores the joint error and may fail in the case where a huge domain gap exists.
> - Our proposal can leverage pseudo predictions while MCD assumes an identical hypothesis space constraint for the two classifiers.
> - MCD uses $L_1$ norm to measure the dissimilarity between the hypotheses while we propose cross margin discrepancy which is theoretically supported by margin theory and can be interpreted with CGAN.
>
> The advantage of our proposal is supported by experimental results in Tab.2, Tab.3 where our method outperforms MCD by __9.7% and 6.7%__ on the average accuracy of VisDA and OfficeHome dataset respectively. Besides, the visualization of feature space with t-SNE showed in Fig.7 implies that our method performs much better in aligning distributions. These improvements cannot be achieved by simply adding a classifier into MCD.

---

### Author Response · Authors · 2022-10-18
**Revisions based on Reviews**

- We modify some math illustrations.

- We move A.4 to the main body Sec.3.3, which shows the comparisons with other methods in an algorithmic scope and demonstrates the generality of our proposal.

- We add Sec.3.5 in the main body to describe the training procedure.

- We add A.6 to show that our algorithm is robust against imbalance label distribution.

- We move the Ablation Study to A.7 due to the page limit.

---

### Decision · Action_Editors · 2022-11-28

**Recommendation:** Reject

**Comment:**

Thanks for your submission to TMLR.

All three reviewers noted concerns about the writing/clarity of the paper (for example, saying it was difficult to follow and the mathematical concepts were not clearly illustrated).  Two of three reviewers still felt this way even after going back-and-forth with authors multiple times.

It seems that the paper needs some additional work still, in particular a heavy rewrite of the technical sections would significantly benefit the paper.  If "major revision" was an option, that's what I would choose here, but that is not an option so I'm recommending reject for this paper.  I would definitely encourage resubmission, as the ideas are solid and the reviewers all felt the paper could eventually be accepted with some additional work.

**Audience:**

Yes, this is very relevant to the TMLR audience.

**Claims And Evidence:**

In general, yes, though there were several issues about clarity of the evidence, as raised by all three reviewers.

---

> ### Author Response · Authors · 2022-11-30
> **Reply to Action Editors**
>
> Thank you for your comments and we sincerely appreciate the hard work and dedication of you and all the reviewers.
>
> We are willing to resubmit, but we hope to at least make sure of what kind of major revision you are expecting.
>
> Since there is no response from the two reviewers after we made a revision based on the requests  including math illustrations, an algorithm flow, a demo code and an additional experiment, we are not sure which part is unclear.
>
> Therefore, if the reviewers still have concerns about the clarity,  could you please tell us some details so that we can revise the paper based on the suggestions.
>
> Besides, when resubmitting, would you mind if we choose you as the AE again?